# A Review on Molecular Docking on HDAC Isoforms: Novel Tool for Designing Selective Inhibitors

**DOI:** 10.3390/ph16121639

**Published:** 2023-11-22

**Authors:** Aliki Drakontaeidi, Eleni Pontiki

**Affiliations:** Department of Pharmaceutical Chemistry, School of Pharmacy, Faculty of Health Sciences, Aristotle University of Thessaloniki, 54124 Thessaloniki, Greece; alikdrak@pharm.auth.gr

**Keywords:** histone deacetylases, molecular docking, isoforms, interactions, enzyme inhibition

## Abstract

Research into histone deacetylases (HDACs) has experienced a remarkable surge in recent years. These enzymes are key regulators of several fundamental biological processes, often associated with severe and potentially fatal diseases. Inhibition of their activity represents a promising therapeutic approach and a prospective strategy for the development of new therapeutic agents. A critical aspect of their inhibition is to achieve selectivity in terms of enzyme isoforms, which is essential to improve treatment efficacy while reducing undesirable pleiotropic effects. The development of computational chemistry tools, particularly molecular docking, is greatly enhancing the precision of designing molecules with inherent potential for specific activity. Therefore, it was considered necessary to review the molecular docking studies conducted on the major isozymes of the enzyme in order to identify the specific interactions associated with each selective HDAC inhibitor. In particular, the most critical isozymes of HDAC (1, 2, 3, 6, and 8) have been thoroughly investigated within the scope of this review.

## 1. Introduction

### 1.1. Introduction to HDACs

Histone deacetylases (HDACs) are a family of proteins that play a major role in the regulation of epigenetic mechanisms of gene expression [1,2,3]. Their dysfunction has been implicated in the development of several different types of cancer. HDAC protein family s normally removes acetyl groups from the amino acid lysine side chain of both histone and non-histone proteins, whilst acetyltransferases (HATs) carry out the reverse process [4,5,6,7,8]. Chaperones, heat shock proteins, nuclear receptors, and certain transcription factors, such as nuclear factor-κB, are among the proteins affected by these enzymes [9,10,11]. After deacetylation, the positively charged *N*-terminal residues of amino acids interact with DNA phosphate groups. This interaction leads to chromatin adopting a compact and stable structure with low transcriptional activity. The action of HAT results in opposite effects, and thus, maintaining an active balance between the actions of two protein families is crucial. Dysregulation of this balance can lead to a variety of diseases, including neurodegenerative and cardiovascular disorders, autoimmune diseases, metabolic disorders, diabetes, and cancer [11,12,13,14,15,16,17]. It is widely recognized that cancer is not solely caused by mutations in DNA but also by the dysregulation of epigenetic mechanisms. These mechanisms encompass the modulation of histone structure, which involves reversible post-translational modifications. Aberrant histone acetylation, resulting in disrupted transcription, is now acknowledged as a hallmark of numerous types of cancer [7,18,19,20]. HDACs are not only implicated in the pathogenesis of cancer but also in its spread and metastasis [21].

In eukaryotic organisms, HDACs comprise 18 enzymes that are categorized into classes based on their structure, catalytic activity, subcellular localization, and homology. HDACs are further divided into specific subclasses. Class I HDACs are subdivided into class Ia and class Ib. Class Ia includes the isoforms HDAC1 and HDAC2, while class Ib includes HDAC3 and HDAC8. Class II is subdivided into IIa, which contains isoforms 4, 5, 7, and 9, and subclass IIb, which consists of isoforms 6 and 10. Class IV includes only the isoform HDAC11. In all of the aforementioned categories, the removal of acetyl groups is facilitated through Zn^2+^-dependent hydrolysis. The zinc ion is bound to a specific region in the enzyme, which is shared among all three classes and consists of one histidine and two aspartic acid residues. Class III HDACs are (NAD)^+^-dependent and exhibit homology to the yeast protein Sir2, which led to their alternative name: sirtuins [11,22,23,24,25]. Classes I, II, and IV show structural and sequence similarities in the catalytic domain. Class III enzymes, on the other hand, show no homology with any other class of enzyme [26,27].

### 1.2. HDAC Inhibition and Inhibitors

The enzymatic activity of HDACs is subject to regulation by a multitude of factors, primarily encompassing post-translational modifications. Concurrently, their activity is impeded by small molecules referred to as HDAC inhibitors (HDACi). These inhibitors are the focus of ongoing research endeavors aiming to devise and synthesize novel therapeutic agents [5]. Two of the most widely known inhibitors are the anticancer drug vorinostat—chemically known as suberoylanilide hydroxamic acid (SAHA)—and trichostatin A (TSA) (Figure 1). These two molecules are structurally similar and lead to a range of effects including apoptosis, autophagy, oxidative stress, and inhibition of angiogenesis, a hallmark of cancer [28,29,30,31]. These compounds inhibit the enzyme through their hydroxamic acid group, which forms a chelating complex with the Zn^2+^ ion of the active site. It should be emphasized that these compounds do not inhibit class III enzymes [32].

Additional inhibitors that have received FDA approval include belinostat and panobinostat, both of which are hydroxamate derivatives; romidepsin, a cyclic peptide; and clidamide, classified as a benzamide (Figure 2). These medications are specifically indicated for the treatment of T-cell lymphoma, myeloma, as well as various solid and hematological cancers [33,34,35,36]. Generally, when HDACs are inhibited, there is an elevation in the level of acetylated histones within the nucleus. This, in turn, triggers the activation of certain regulatory genes in cancer cells that were previously silenced [37]. In this way, inhibitors exert control over proangiogenic genes, prevent angiogenesis, and lead to apoptosis and the death of mitotic cells [37,38,39]. It is important to acknowledge that normal cells exhibit resistance to the action of HDACis [40]. Certain inhibitors are currently being studied for their potential therapeutic effects on neurological disorders, including Huntington’s disease [41,42]. Further diseases that may be treated by HDACis include HIV, diabetes, inflammatory diseases, and parasitic diseases that infect humans, such as schistosomiasis and trypanosomiasis [43,44].

In addition to their therapeutic effects, the above-mentioned inhibitors also present a number of side effects, such as thrombocytopenia, neutropenia, fatigue, and prolongation of the QT interval. This is one of the reasons why research is turning to the study of plant-derived inhibitors such as luteolin and apigenin, compounds that chemically belong to the flavones, a subgroup of the flavonoids (Figure 3) [5,29,37,45,46,47,48,49]. The compounds listed above appear to act on the same targets as HDAC inhibitors [45,50,51]. Apigenin interferes with HDAC function and demonstrate anti-proliferative effects in breast cancer cells [52]. Luteolin exhibits an inhibitory effect on epithelioid cancer cells [51]. Simultaneously, the fungal metabolite apicidin demonstrates inhibitory activity against HDAC enzymes [53].

All HDACis share a common pharmacophoric structure. They consist of a group that forms a chelating bond with the Zn^2+^ ion, a linker group characterized by hydrophobicity and a capping group capable of interacting with hydrophobic regions of the enzyme as well as with amino acids in the side-chains of the peripheral domains of the active site (Figure 4). The latter is the most important as it confers selectivity for the isoform that inhibits the enzyme [54,55,56,57,58]. It is important to emphasize that while it exists a general category of chemical compounds that inhibit HDACs, there are numerous other compounds with inhibitory activity where distinguishing the three mentioned areas is not straightforward [59].

In conclusion, HDACs play a pivotal role in regulating epigenetic mechanisms, rendering them a promising target for the development of anticancer and other pharmaceutical agents [22]. As expected, extensive research efforts have been directed towards the development of novel HDACIs that exhibit improved selectivity and reduced toxicity [28]. Moreover, the therapeutic efficacy of these compounds is strongly associated with their selectivity against specific isozymes of the enzyme [60,61]. It is therefore imperative to study the binding mechanisms of each inhibitor to the different isozymes of the enzyme, in order to allow for the development of more selective and effective derivatives [62].

## 2. Molecular Docking Studies on HDACs

HDAC enzymes are well-known for their functional role mediated by the active site, which encompasses an 11 Å binding site housing a zinc ion (Zn^2+^) serving as a cofactor. Their mechanism of action involves the activation of bound water molecules for nucleophilic attack and subsequent hydrolysis [63,64]. At the same time, a tetrahedral zinc intermediate alkoxide is formed, which is stabilized by enzymatic residues and releases the acetate and lysine residues of the target protein as reaction products. Moreover, an inner tubular cavity, approximately 14 Å deep, is situated beneath the active site. It has been suggested that this cavity serves as the exit pathway for the acetate residue [63]. First, crystallographic structures of human HDACs complexed with their inhibitors reveal HDACIs’ mechanism of action through binding to active channels and stereochemical inhibition of substrate hydrolysis [65].

In this review, an attempt was made to clarify the interactions of several selective inhibitors of the most critical isozymes of HDAC (1, 2, 3, 6, and 8). The development of computational chemistry tools, particularly molecular docking, has been proven to be a fast and inexpensive technique enhancing the precision of designing molecules with inherent potential for specific activities. Molecular docking is not a panacea. The main weaknesses considering unsolved are (a) the flexibility of the protein (dynamic aspects of protein ligand binding are ignored when using a native protein [66]) and (b) the presence or absence of water molecules in the binding site (recognize constitutional waters important for the binding [67,68,69]). To tackle the first problem, molecular dynamics simulations (MD) of the protein are performed exploring the different protein conformers [70], and MD simulations are performed for the final complexes (ligand-protein) [70] Regarding the second problem, it is important to recognize and include only key waters in the study and apply software such as GLexX or GOLD [67]. This problem has effective solutions available. So, which is the answer to the question of medicinal chemists “When should I use docking?”. Docking can be applied to the virtual screening of compound libraries, drug design, and lead optimization. Irwin, J.J. and Shoichet, B.K. [71] propose good strategies leading to successful drug candidates applying docking studies and other screening techniques to medicinal chemistry programs. Moreover, novel advancements in the four main aspects of molecular docking approaches have been made: (i) available benchmarking sets for pose prediction, binding affinity, and virtual screening; (ii) exploration of the chemical space (fragment-based drug design); (iii) advances in consensus methods; and (iv) artificial intelligence–machine learning algorithms [72]. Thus, molecular docking can be considered a valuable tool for drug design, especially when it is supported with MD studies or when its results are confirmed by biological experiments. Additionally, nowadays, the application of molecular docking has expanded to the prediction of adverse effects, polypharmacology, drug repurposing, and target fishing and profiling [73].

Therefore, it was considered necessary to review the molecular docking studies conducted on the major isozymes of HDAC in order to identify the specific interactions associated with each selective isozyme HDAC inhibitor. All of the studies included in this review are supported by MD simulations or biological results.

### 2.1. HDAC 1

Isozyme 1, a member of the first class of HDAC enzymes, plays a key role in cell proliferation [22,74]. HDAC 1 has been implicated in the pathogenesis of various cancer types, such as gastric and prostate cancer. Furthermore, it appears to exert an influence on the progression of breast cancer by modulating both the expression and the transcriptional activity of the estrogen receptor protein alpha [75,76,77]. It is noteworthy that the 14 Å tunnel plays a crucial role not only in the catalytic activity of HDAC enzymes but also in the context of specific inhibition [78]. Hence, computational tools like molecular docking play a vital role in the design and development of novel drugs that can selectively target and inhibit HDAC1 [79,80,81,82].

Scafuri, B. et al. [83] conducted a study comparing the binding modes of flavone, apigenin, and luteolin (Figure 3) to the HDAC1 enzyme (Figure 5) with that of vorinostat. The highest resolution HDAC1 protein has been selected (PDB ID: 4BKX), being in complex with the dimeric metastasis-associated protein 1 from the nucleosome remodeling, and then the deacetylase (NuRD) complex was analyzed. Docking simulations were performed using AutoDock 4.2. Molecular docking was performed after removing the dimer and customizing three binding systems: one without water molecules, one with two water molecules in the binding site, and one with water molecules bound to the ligand to explore potential binding facilitation. Both blind docking and focused docking, limited to the binding site, were carried out. It has to be noted that, as a validation method, cluster analysis was performed on the docking results with an RMSD tolerance of 2 Å using the initial coordinates of the reference structure—vorinostat. After cluster analysis, only the results with the most favorable binding energy were retained. The initial molecular docking study of vorinostat demonstrated an interaction energy of −8.46 Kcal/mol. Notably, the orientation of the molecule allowed the oxygen attached to the C8 carbon to interact with the zinc ion as expected [28]. Interestingly, in blind docking, flavone and apigenin exhibited more favorable binding energies compared to the reference compound, while luteolin exhibited a less favorable binding energy [73]. It is important to note that while vorinostat interacts with the zinc anion through two oxygen groups (with different interactions for HDAC1 and HDAC2), the compounds being investigated interact with the zinc anion through a carbonyl group. Specifically, the carbonyl group on the C4 carbon of flavones mimics the carbonyl of the acetyl group found on acetylated lysine, which serves as a substrate for the enzyme. Additionally, the aromatic structure of these molecules hinders the activation of water molecules and nucleophilic attacks [84]. The molecular docking results revealed that flavone’s binding to HDAC1 is strengthened by aromatic interactions with the amino acid TYR303. Moreover, its B-ring interacts with phenylalanine residues (PHE150 and PHE205), leading to a conformational change that blocks both the entry into the binding site and water entry. In addition, all of the compounds investigated interact with the amino acids GLY149 and GLY301 through van der Waals forces. The presence of GLY149, situated at the entrance of the enzyme’s binding pocket between the two preceding phenylalanines (PHE150 and PHE205), significantly contributes to the final shape of the channel. The compounds also engage in interactions with ASP176/181 (via van der Waals interactions for flavones and via hydrogen bonding for luteolin and apigenin), HIS178 (through π-interactions), and ASP264 (specifically observed in apigenin). The coupling of these three amino acids with the compounds is particularly significant due to their coordination with the zinc anion. Additionally, interactions occur with HIS140 (via hydrogen bonding), HIS141 (through π-cationic bonding), and TYR303 (involving lone pair π interactions). It is noteworthy that the last three amino acids play a crucial role in the deacetylation mechanism, with the histidines being involved in charge transfer to the active site. The reference compound, vorinostat, also interacts with these amino acids. Lastly, the amino acids MET30, LEU139, CYS151, and GLY300 exhibit interactions without an apparent role.

Sixto-López, Y. et al. [85] conducted molecular dynamic simulation and a molecular docking study to examine the binding of various inhibitors to both the native structure of HDAC and the structure of HDAC1 with retained K^+^ ions. The objective was to investigate whether the presence or absence of potassium ions impacts co-receptor binding. The enzyme structures were obtained after performing molecular dynamics (MD) simulations and analyzing root-mean-square deviation (RMSD) data. The chemical compounds chosen for the study (Figure 6) included dacinostat and quisinostat, which are hydroxamate derivatives, the cyclic peptide chlamydocin with an epoxyketone group, and a benzamide compound (AC1OCG0B). These compounds were compared to vorinostat (SAHA), which was utilized as the reference compound due to its pan-inhibitory characteristics. Vorinostat exhibits the ability to inhibit all isozymes of HDAC without displaying selectivity. All of these derivatives have been reported in the literature as potent HDAC inhibitors (vorinostat CI_50_ (nM) = 61.8 [86]; dacinostat CI_50_ (nM) = 32 [87]; AC1OCG0B CI_50_ (nM) = 7 [81] chlamydocin CI_50_ (nM) = 0.11 [88]; quisinostat CI_50_ (nM) = 0.1).

For the docking studies Autodock 4.2 [89] was used along with AutoDock4Zn, an improved force field of AutoDock for the docking of zinc metalloprotein [90]. HDAC1 (PDB ID: 4BKX) was selected for the docking. MD simulations revealed the most populated conformation according to the RMSD analysis. It was observed that all compounds, except for chlamydocin, successfully reached the catalytic center of the enzyme in its native form. Chlamydocin has a cyclic peptide capping group that interacts with the amino acids ASP99, PHE150, LYS200, TYR204, PHE205, ARG270, LEU271, and CYS273, thus blocking entry into the catalytic site. In the crystalized enzyme, there are two K^+^ ions, which were retained, and a specific Zn^2+^ ion forming the catalytic center of the enzyme. For the other two compounds, the benzamide and hydroxamate groups did not directly contribute to their binding to the Zn^2+^ ion. Instead, they formed π-interactions with the Zn^2+^ ion through the phenyl ring positioned adjacent to a carbonyl group. In the enzyme structure where K^+^ ions were removed, all compounds except chlamydocin successfully reached the catalytic center due to the aforementioned blocking effect of its cyclic peptide capping group. In the enzyme form where potassium is only retained at site 1, significant changes occur. Dacinostat and quisinostat undergo reorientation, no longer interacting through their hydroxamate groups. Instead, they flip and are oriented away from the potassium ions. Chlamydocin, quisinostat, and dacinostat establish π-cation interactions with zinc through their phenyl ring. On the other hand, vorinostat is the only compound that enters site 1 along with the potassium ion, exhibiting interaction with zinc through its hydroxamate group. In the enzyme form where potassium is only retained at site 2, all compounds, except chlamydocin, interact with zinc ions via their hydroxamate or benzamide groups. Chlamydocin engages with PRO29, GLY149, and PHE205, as well as the amino acids HIS178 and ASP264 of the catalytic center through its cyclopeptide moiety, while its epoxide group is directed towards the channel-releasing acetate group. Similarly, SAHA and the benzamide AC1OCGB orient their phenyl and acetamide groups accordingly. In this case, the protein structure is distorted, but the binding mode of quisinostat and dacinostat molecules remains unchanged.

Silva Urias, B. et al. [91] designed, synthesized, and evaluated a series of resveratrol analogues. Resveratrol exhibits inhibitory activity against HDACs but lacks selectivity towards specific isozymes [92,93]. In the present study, docking studies guided the type of groups, e.g., OH, NH_2_, and pattern (o- or p-) of substitution so as to interact with the enzyme. The researchers investigated the impact of adding an aminobenzamide subgroup to the resveratrol molecule on its activity. A molecular docking analysis was performed using Schrodinger (2019–4) Maestro v12.2, revealing that resveratrol can enter the binding pocket of HDAC1 without directly interacting with the zinc ion, resulting in an interaction energy of −5.565 kcal/mol, confirming previously reported findings [94]. The observed interactions were mainly focused on the active site of the enzyme, including the formation of hydrogen bonds with residue ASP176 and π-interactions with amino acids HIS141 and HIS178 (Figure 7). Upon introducing the 2-aminobenzamide group, the orientation of the molecule was altered, directing it towards the metal (Zn^2+^) side of the active site where interactions took place. The substituted compounds exhibited more consistent interaction energies, better docking scores, and displayed additional hydrogen bonds with amino acids HIS145, HIS146, and GLY154. However, some of the synthesized molecules faced challenges in adapting to the active site due to their rigidity and lack of flexibility. In contrast, certain saturated compounds that possessed greater flexibility were able to achieve a more favorable position within the active site. Among them, the two 2-aminobenzamide derivatives (saturated and unsaturated) presenting the higher docking scores have been further simulated in order to evaluate the reliability of the selected poses in HDAC1. In all of these studies, the RMSD was below 2 Å. Moreover, docking results are in accordance with the initial enzymatic inhibition screening of compounds against HDAC1. Notably, when considering a flexible linker, substituting the carbon atom with either nitrogen or oxygen at the linker site did not appear to have a significant impact on the binding interactions observed.

### 2.2. HDAC 2

HDAC isozyme 2 has been identified to be dysregulated in various types of cancer [32]. It has been observed that HDAC isoform 2 exhibits elevated activity in certain cancer types, including stomach, prostate, colon, and kidney cancers [95,96,97]. Additionally, the HDAC2 isozyme has been implicated in the development of Alzheimer’s disease [16]. The HDAC2 isozyme exhibits a length of 11 Å and features an internal active site tunnel cavity with two PHE residues positioned on opposite sides. Near the base of the tunnel, there are HIS and ASP residues that facilitate coordination with the active site. Positioned at the entrance of the cavity, there is a GLU103 residue on the left, which imparts a more negative charge to this isozyme. On the right side, there is a TYR209 residue [98]. Among the known inhibitors of HDACs, panobinostat appears to exhibit selectivity towards HDAC2 isozyme [99].

Stoddard, S.V. et al. [98] designed a series of hydroxamate derivatives to investigate their molecular binding and selective inhibition of HDAC2 and HDAC8 enzymes using panobinostat as a reference compound. Their scope is to identify important molecular characteristics for selectivity against HDAC2 and consequently proceed to the synthesis of the selected compounds and their biological evaluation. For this research work, HDAC2 crystal structure (PDB ID: 4LXZ) was selected and Surflex-Dock Geom (SFXC) protocol was used for the docking. In their study, they replaced the central aromatic ring of panobinostat with other rings containing heteroatoms (Figure 8). The results revealed that panobinostat’s hydroxamate group coordinated with the zinc ion and formed hydrogen bonds with TYR308. Additionally, hydrogen bonding occurred with the amino acid ASP104 through the nitrogen at the linker site of the compound. The compound also engaged in two parallel π-π interactions with amino acids PHE155 and PHE210. When the central ring of the compound was replaced by a pyrazine ring (compound TOI1), the orientation of the indole moiety of the compound towards the enzyme significantly differed. The polar hydrogen of the indole approached HIS33 and moved away from GLU103, while the conformation of the six-membered ring of the indole approached GLU103. Simultaneously, aromatic interactions with HIS33 and PHE155, as well as hydrogen bonding with TYR209 and HIS183, were observed. This compound also exhibited parallel π-π interactions with PHE155 and PHE210, but the pyrazine ring was flatter compared to panobinostat, allowing for closer approximation to PHE210.

When the central ring of the compound was replaced by a purine ring (ETS5 compound), the hydroxamate group formed hydrogen bonds with TYR308, HIS146, and HIS145. Aromatic interactions with HIS33 and PHE155, as well as hydrogen bonding with GLU103 (through the indole) and ASP104 (through amine of linker), were also observed in this case. The compound also exhibited parallel π-π interactions with PHE155, PHE210, and HIS183. This can be attributed to the larger aromatic surface area of purine, allowing it to penetrate deeper into the enzyme cavity. The ETS4 compound with a quinoxaline ring displayed similar π-π interactions with PHE155 and PHE210, as well as an additional interaction with HIS183. The imidazole (ETS1) and oxazole (ETS2) ring compounds also exhibited π-π interactions with the two PHE residues mentioned earlier, while their indole moieties showed stacking interactions with PHE155 and HIS33. Moreover, their hydroxamate groups not only coordinated with the zinc ion but also formed hydrogen bonds with the amino acids TYR308, HIS145, and HIS146. The indole ring in both molecules was oriented towards the left side of the active site entrance, forming a hydrogen bond with GLU103. ETS2 forms an additional hydrogen bond through the linker amine, while the same amine in ETS1 forms three additional hydrogen bonds with three subunits. The compounds with an isoxazole ring (ETS3) and pyrazole ring (TOI4) also exhibit π-π interactions with the two PHE residues mentioned above. Their hydroxamate group only coordinates with the zinc ion, and the indole ring in both molecules is oriented towards the right side of the active site cavity. Only in the case of ETS3 does hydrogen bonding with PHE210 occur. In both molecules, the amine linker forms hydrogen bonds with ASP104, and in the case of TOI4, there is an additional hydrogen bond with this amino acid through the pyrazole moiety. All of these compounds showed better interaction effects than Panobinostat. It is worth noting that the study confirmed the existence of an aromatic pocket in the HDAC2 isozyme that includes the amino acids HIS33 and PHE155.

Scafuri, B. et al. [83], additionally to HDAC1, conducted a comparative analysis of the binding modes of flavone, apigenin, and luteolin to the HDAC2 enzyme using the same programs as previously reported in comparison to the binding of vorinostat. The reference structure of HDAC2 bound to vorinostat was used to validate the reproducibility of the docking results. The docking simulations between HDAC2 and vorinostat revealed that the interaction occurs between the hydroxyl group of the compound and the Zn^2+^ cofactor of the enzyme. This interaction differs from that observed in HDAC1 but exhibits a similar binding energy. Notably, in the case of HDAC2, flavone and apigenin demonstrated more favorable binding energy compared to the reference compound in blind docking experiments [83]. The compounds under investigation exert their effects on HDAC2 in a similar manner to HDAC1 by mimicking the acetyl group of the lysine residue in the substrate. This structural mimicry allows the compounds to interact with the enzyme’s active site and inhibit its function [84]. The molecular docking results revealed that, like HDAC1, the binding of flavone to HDAC2 (Figure 9) is enhanced by aromatic interactions with the amino acid TYR308. Additionally, the B ring of flavone interacts with phenylalanine residues (PHE155 and PHE210), resulting in a conformation that blocks both entry into the binding site and water access. Similarly, all of the compounds studied exhibit van der Waals interactions with the amino acids GLY154 and GLY306, which play a significant role in the determination and shaping of the entrance of the enzyme binding pocket. Moreover, the compounds interact with other amino acids, including ASP181 (flavone and apigenin through van der Waals forces and luteolin through hydrogen bonds), HIS183 (flavone through π-interactions, apigenin through carbon–hydrogen bonds, and luteolin through van der Waals forces), and ASP264 (only in apigenin through van der Waals forces). Coupling with these three amino acids is particularly important because of their association with the zinc anion. Furthermore, there are interactions observed with HIS145, HIS146, and TYR303. HIS145 engages in apigenin π-cationic interaction, while flavone and luteolin form hydrogen bonds with it. Moreover, HIS146 participates in flavone π-cationic interactions, apigenin forms hydrogen bonds, and luteolin exhibits π-interactions with it. Finally, TYR303 is involved in lone-pair π-interactions. It is important to note that these last three amino acids play a crucial role in the deacetylation mechanism, particularly histidines, which are involved in charge transfer to the active site. The reference compound, vorinostat, interacts with them. Finally, the amino acids ASP104 and GLN265 interact without an apparent functional role.

Silva Urias, B. et al. [91] conducted a study investigating the effects of adding an aminobenzamide subgroup to the resveratrol molecule, as mentioned previously. Molecular docking analysis was performed to examine the binding of resveratrol to the HDAC2 (Figure 10) enzyme (PDB ID: 4LY1) using Schrodinger (2019–4) Maestro v12.2. The designed derivatives have been synthesized and further evaluated. Biological results revealing the most active derivative against HDAC confirm the docking studies results. The results revealed that resveratrol enters the HDAC2 binding pocket without interacting with the zinc ion, exhibiting an interaction energy of −6.453 Kcal/mol. The detected interactions involve the enzyme’s binding pocket, including hydrogen bonding with residue GLY143 and π-interactions with amino acid PHE155 (Figure 10). Upon the introduction of the 2-aminobenzamide group, the orientation of the molecule is altered, directing it towards the metal (Zn^2+^) side where an interaction takes place. The substituted compounds exhibited more favorable binding energies and additional hydrogen bonds with amino acids HIS145, HIS146, and GLY154. It should be noted that some of the synthesized molecules showed limited adaptability to the active site due to their rigidity caused by unsaturation [91]. Of course, this happened in a different way than in HDAC1 because of the wider input found in the active center of HDAC2 [100]. In contrast, the flexibility of certain saturated compounds facilitated their adoption of a more favorable position within the active site, resulting in more favorable interaction energies. Additionally, the saturated analogues formed additional hydrogen bonds with amino acids LEU276 and GLU208. Notably, when a flexible linker was present, changing the carbon atom to nitrogen or oxygen at the linker site did not yield any significant differences. On the other hand, the introduction of polar substituents, such as hydroxy, amino, and amido groups, at the cap of the molecule promoted hydrogen bonding interactions with LEU276.

In their study, Mourad, A. et al. [101] synthesized and biologically evaluated α-phthalimido substituted chalcones (Figure 11) with the aim of achieving anticancer activity by inhibiting both tubulin and HDAC1/2. The biological experiments on HDAC2 revealed that the 3,4,5-trimethoxyphenyl derivative presented stronger inhibitory activity than entinostat. Additionally, a molecular docking study, focusing on the HDAC2 isoform and using entinostat as a reference compound, was conducted for the three most active compounds. The MOE program was used for the docking studies of the novel derivatives into HADAC2 (PDB ID: 4LXZ). Among the compounds tested, the one with a 3,4,5-trimethoxyphenyl substituent, being the stronger biologically active derivative, exhibited the strongest binding energy of −9.01 kcal/mol. It formed hydrogen bonds with amino acids ARG197 and ASN331 via the oxygen of the chalcone group and the phthalimido group. The compound with a 2,4-dimethoxyphenyl substituent showed the second-best binding energy and established hydrogen bonds with amino acid ASN331 through the chalcone oxygen, as well as π-interactions with ARG311 through the aromatic phthalimido ring. The compound with a substituted p-methoxyphenyl group displayed the third-best binding energy and demonstrated hydrogen bonding with ASN331. The reference compound (entinostat) demonstrated similar bonding patterns along with a π-hydrogen interaction with PRO344.

### 2.3. HDAC 3

The HDAC3 isozyme is of particular interest because it has been implicated in transcriptional repression in several types of cancer [102]. HDAC3 has the ability to form a complex with nuclear receptor co-repressor (N-CoR) and other receptors, such as thyroid receptor, and through these complexes can participate in transcriptional repression. In particular, forming such complexes of the enzyme leads to increasing its activity [103,104,105]. As with the previous isozymes, this isoform has been implicated through epigenetic alterations in several types of cancer, cardiovascular disease, neurodegenerative disease, and memory and learning disorders [106,107,108,109,110]. HDAC isozyme 3 has been demonstrated to exert a negative regulatory effect on long-term memory processes. Conversely, inhibition of this isozyme shows promise in contributing positively to various cognitive disorders. Moreover, there is evidence suggesting that HDAC3 inhibitors may hold potential for combating drug addiction [106,111,112]. Furthermore, studies have demonstrated the involvement of HDAC3 in the regulation of the B7-H1 gene, resulting in increased levels of interferon-γ (INF-γ) in gastric cancer. Additionally, overexpression of the secreted isozyme of HDAC3 has been associated with reduced expression of p21 in colorectal cancer [113,114], and HDAC3 has been found to play a role in reducing cell proliferation in breast cancer through its interaction with the CREB protein. It also plays a regulatory role in other important genes such as HIF-1α, NF-κB, and STAT3 [115,116,117].

The HDAC3 enzyme is distinguished by its unique C-terminal domain, setting it apart from other enzymes in the HDAC family. Several amino acids within the active site, including TYR198, ASP92, PHE199, and TYR107, play a crucial role in its activity towards specific substrates [103,104,105]. Additionally, HDAC3 possesses an internal cavity near the binding site that accommodates the presence of Zn^2+^ and facilitates specific substrate binding [36]. An important structural characteristic of this particular isozyme lies in the unique arrangement of amino acids. Specifically, the side chain of amino acid TYR107 exerts pressure on the neighboring residue LEU133, creating a small pocket that is notably smaller than those observed in other isozymes. As a consequence, this specific structural feature restricts the binding capacity of bulky inhibitors to the HDAC3 enzyme [118].

Routholla, G. et al. [119] synthesized several benzamide derivatives containing aryl or heteroaryl groups, wherein the linker group was absent. The compounds were evaluated for their ability to inhibit HDAC1, HDAC2, HDAC3, HDAC6, and HADAC8 isozymes. The *N*-(2-aminophenyl) quinolone-6-carboxamide and the *N*-(2-amino phenyl)-*1H*-indole-6-carboxamide displayed the best IC_50_ values of 0.560 µM and 2.077 M to HDAC3 compared to the other isozymes. Moreover, the reference compound CI994 (IC_50_ = 0.902 µM) was nonselective towards HDAC3 over HDAC1 and HDAC2. However, *N*-(2-aminophenyl) quinolone-6-carboxamide, besides being a very potent HDAC3 inhibitor, displayed 46-fold selectivity for HDAC3 over HDAC2 and 33-fold selectivity over HDAC1. The *N*-(2-amino phenyl)-*1H*-indole-6-carboxamide, as previously mentioned, is 4 times less active to HDAC3 than the *N*-(2-aminophenyl) quinolone-6-carboxamide but still retained a minimum 5-fold HDAC3 selectivity over other isozymes tested (HDAC1 and HDAC2). Finally, these two compounds and the reference one have been studied for their inhibitory activity against HDAC6 and HDAC8 to determine their selectivity against HDAC3. 

Thus, molecular docking was performed on these two derivatives, which demonstrated better selectivity for the HDAC3 isozyme, as well as on a reference compound, CI994 (Figure 12). The Glide module of the Schrodinger Maestro software [120] was used with HDAC3 isozyme (PDB ID: 4A69). The study revealed that all compounds exhibited a nearly parallel orientation upon binding to the enzyme’s active site (Figure 13). The carbonyl group of the benzamide formed hydrogen bonds with the amino acid TYR298, while the -NH group of the benzamide formed hydrogen bonds with the amino acid GLY143. Additionally, π-π stacking interactions were observed with the amino acid PHE144. Interestingly, the study concluded that benzamide derivative with a quinoline group at the 6-position exhibited more favorable binding to the HDAC3 isozyme compared to a compound containing indole. Based on this study *N*-(2-aminophenyl) quinolone-6-carboxamide can be considered within the series as a selective lead molecule of HDAC3.

Kumbhar, N. et al. [121] conducted a study focusing on the significance of HDAC isozyme 3 in various malignancies. They employed computational tools to screen novel compounds with selective inhibitory activity against HDAC3. After performing pharmacophore modelling, virtual screening of large compound libraries, and drug-likeness filtration, 174 hit compounds were identified. These 174 screened hit compounds and training set compounds along with a known inhibitor compound, TSA (the most active compound within the training set) (Figure 1), underwent molecular docking and molecular simulation studies using the GOLD software program v5.2.2 revealing three top-scored hit compounds (Figure 14). These three top-scored compounds were analyzed for their binding mode to HDAC3 (PDB ID: 4A69) (Figure 15). TSA showed hydrogen bonding interactions with the amino acids ASP91, HIS171, PRO200, and GLY295, and coordinated with the Zn^2+^ ion. The Hit1 compound displayed hydrogen bonding interactions with the amino acids MET23, HIS133, GLY142, and PHE199, all of which are located in the active site of the enzyme. Additionally, this compound coordinated with the Zn^2+^ ion. Notably, Hit1 also exhibited π-π interactions with the amino acids PRO22 and LEU265 (π-alkyl type) and with ASP92 (π-lone type). Furthermore, its fluorine moiety demonstrated numerous weak interactions. These hydrophobic interactions seemed to stabilize the structure of the compound within the active site of the enzyme. Compound Hit2 engages in a bifurcated hydrogen bond with amino acid PHE199 and forms weak hydrogen bonds with HIS171, PHE198, and PRO200, which are amino acids in the catalytic region of the enzyme. Additionally, Hit2 can coordinate with the Zn^2+^ ion and establishes two π-π interactions, specifically π-stacking type interactions with HIS133 and HIS134 residues. On the other hand, the Hit3 compound also demonstrates the ability to coordinate with zinc and exhibits π-π-alkyl-type interactions with HIS133, HIS134, and PRO200. Overall, the interactions observed in these three compounds are quite similar to those found in TSA, suggesting that further investigation of these compounds for their selective inhibition of HDAC3 could be promising. The obtained results are analogues with earlier crystallographic data of FDA-approved drugs which co-crystallized with HDACs.

Bülbül, E.F. et al. [122] conducted a comprehensive analysis using cheminformatics, computational chemistry, and molecular docking techniques to investigate the binding mechanisms of compounds containing an *N*-alkylhydrazide group with the HDAC3 enzyme. Their investigation involved a library of compounds from existing literature with available inhibitory activity data for each compound. Their research was carried out using a library of compounds derived from the literature with evaluated inhibitory activity. Compounds were subject to docking studies using Glide implemented in Schrödinger Suite [123] and the X-ray crystal structure of HDAC3 (PDB ID: 4A69). For 6-methoxy-N-(4-(2-propylhydrazine-1-carbonyl)benzyl)benzofuran-2-carboxamide and 4-(((2-(1H-indol-2-yl)ethyl)amino)methyl)-N′-propylbenzohydrazide (Compound **1** and **2**) (Figure 16), the mode of binding was further analyzed (Figure 17), and it was found that the hydrazide group plays an important role as it coordinates with zinc in two ways: first, through the nitrogen of the group, and second, through the oxygen of the carbonyl. Additionally, the hydrazide group engaged in crucial hydrogen bonding with specific amino acids, namely HIS134, HIS135, and TYR298, situated at the bottom of the catalytic site. Moreover, the linker group of these compounds, characterized by aromaticity, effectively accessed the hydrophobic tunnel of the enzyme, forming favorable π-π interactions with PHE144 and PHE200. Furthermore, the cap group of the compounds established hydrogen bonding interactions with ASP93 and hydrophobic interactions with HIS22 and PPRO23. The docking results demonstrated that when the *N*-alkyl chain attached to the hydrazide group was a propyl or butyl chain, the compounds displayed potent inhibitory activity against the enzyme. Conversely, when the chain was longer, such as in the case of the pentyl chain, the inhibitory activity was significantly reduced. The above results confirm the existence of a smaller foot pocket of the enzyme, as mentioned above.

### 2.4. HDAC 6

The HDAC6 enzyme is classified under class IIb of HDACs, and it possesses a distinct characteristic of deacetylating other non-histone substrates of significant importance, including tubulin and p53 [124,125]. It is important to note that HDAC6 is primarily located in the cytoplasm, in contrast to most other HDAC isozymes, which are mainly found in the nucleus [126]. HDAC6 contains two catalytic deacetylation domains known as DD1 and DD2 alongside several other domains that serve specific functions [127,128]. A distinctive structural feature of the DD2 catalytic domain of HDAC6 is the presence of a wide rim, enabling the enzyme to accommodate and be inhibited by bulky and predominantly aromatic compounds [54,126]. Indeed, HDAC6 has been implicated in various types of cancer and their metastases, including the metastasis of Burkitt’s lymphoma cells. Therefore, selective inhibition of this isozyme holds promise as a potential anti-cancer therapy [129,130,131]. At the same time, it is crucial to acknowledge the pivotal role of this isozyme in the pathophysiology of Alzheimer’s disease. The presence of acetylated tau proteins has been linked to ameliorations in the disease’s pathology, underscoring the significance of HDAC6 inhibition as a potential therapeutic strategy [132,133,134,135]. Certainly, the selectivity of HDAC6 inhibitors has emerged as a significant advantage, as they demonstrate reduced cytotoxicity to mammalian cells and fewer pleiotropic reactions compared to non-selective pan-inhibitors. This is a beneficial property that makes selective HDAC6 inhibitors a more encouraging and safer therapeutic option [136,137,138].

Zeb, A. et al. [139] undertook a study to identify compounds with selective HDAC6 inhibition potential aimed at disrupting the pathogenesis of Alzheimer’s disease. Employing computational tools, they meticulously screened an extensive compound database. Their approach encompassed the generation of a pharmacophore model based on desired inhibitor structural attributes, comprising a hydrogen bond acceptor atom, two hydrogen bond donor atoms, and two hydrophobic moieties. Subsequent virtual screening of the compound pool, complemented by RMSD and cluster analyses, yielded 841 drug-like molecules. Molecular docking studies through GOLD v5.2.2. software to HDAC6 (PDB ID: 5EDU) revealed 11 molecules as candidates while molecular dynamics simulation assessed the pose orientation and binding mode under physiological conditions of the final three promising hit compounds. Trichostatin A (Figure 1) was used as a reference compound. This investigation notably underscored the significance of polar interactions between the compounds and specific amino acids within the enzyme’s active site, notably HIS610, HIS611, ASP649, HIS651, and TYR782, in the inhibition process. HIS610, HIS611, and the active site’s zinc ion collectively confer a positive charge, necessitating the inhibitor to bear a corresponding negative charge to facilitate polar interactions. In concurrence with the molecular docking outcomes, the reference compound TSA displayed polar interactions with HIS610 via its carbonyl nitrogen as well as with ASP742 through its amino group. Further contributing to the inhibitory mechanism, the compound’s carbonyl coordinated with the Zn^2+^ ion. Additionally, the methyl groups of the compound engaged in hydrophobic interactions with HIS611, HIS651, PHE680, and LEU749 amino acids.

Concerning the investigated hit compounds (Figure 18), all three displayed the same orientation and binding within the enzyme’s structure. Notably, the carbonyl oxygen of the first compound (glycodeoxycholic acid) establishes a hydrogen bond with the amino acid HIS610, concurrently engaging in an electrostatic interaction with the Zn^2+^ ion. Additional hydrogen bonds are formed with amino acids ASN654, PRO681, and TYR782. Meanwhile, the steroid core of the molecule engages in hydrophobic interactions with CYS618, PHE680, and PRO681. The second compound established hydrogen bonds with the nitrogen of amino acid HIS611 and the side-chain oxygen of amino acid ASP649. Additional hydrogen bonding occurred with amino acids GLY619 and GLY780. Concurrently, hydrophobic interactions were evident with amino acids PRO501, HIS651, PHE680, LEU749, and TYR782. Furthermore, this compound demonstrated diverse van der Waals interactions across distinct sites within the enzyme’s active site. In the case of the third compound, characterized by an acetic acid residue, its carbonyl group formed a hydrogen bond with amino acid HIS611, concurrently coordinating with the Zn^2+^ ion within the active site. Additional hydrogen bonds were established with amino acids HIS651 and PRO681. Notably, this compound engaged in several hydrophobic and van der Waals interactions with multiple residues within HDAC6’s active site.

Since several HDAC inhibitors are designed to mimic acetyl lysine, the endogenous substrate, Song, Y. et al. [140] have tried to design and synthesize a series of anthraquinone-based HDAC6 inhibitors. As shown in Figure 4 HDAC inhibitors consist of a group chelating Zn^2+^ ion, a linker group characterized by hydrophobicity and a capping group. Hydroxamic acid is the most common zinc chelating group [141]. Song, Y. et al. [140] in this study performed virtual screening of a multitude of structures with binding potential to the HDAC6 enzyme through the cap and linker groups to design selective inhibitors by adding zinc-binding groups to the main structures. They have selected from the ZINC15 [141] database 445 compounds bearing the ring subset of drugs and filtered macrocyclic compounds, leaving 235 compounds for docking on HDAC6 (PDB ID: 5EF7). After calculating the binding affinities of different structures through Autodock Vina, they concluded that the three-ring structures appeared to be promising. The study resulted in the structure of anthraquinone (Figure 19), which demonstrates the ability to bind to the binding pocket of the enzyme. The two hydroxy groups of anthraquinone are involved in hydrogen bonds with amino acids SER531 and HIS614, while one phenyl ring is directed towards the hydrophobic channel of the enzyme, forming π-π interactions with amino acids PHE643 and PHE583. New compounds were then designed and synthesized by adding a hydroxamic group to the anthraquinone structure in order to interact with the zinc ion. The *N*-hydroxy-9,10-dioxo-9,10-dihydroanthracene-2-carboxamide (Figure 20) with the most efficient and selective activity against HDAC6 (IC_50_ = 56 nM) compared to the reference compound vorinostat (IC_50_ = 226 nM) was further investigated for its drug likeness and selectivity to HDAC6 over the other isozymes. The carbonyl and hydroxyl of its hydroxamic group showed interaction with the zinc ion, while these two groups and the amine of the hydroxamic group showed hydrogen bonding with the amino acids TYR745, HIS573, and GLY582. Conversely, the phenyl rings alongside the intermediate quinone ring establish π-π interactions within the hydrophobic channel of the enzyme. Additionally, the carbonyl groups of the compound engage in hydrogen bonding interactions—one with the amino acid HIS614 and the other with the hydroxyl moiety of SER531. Remarkably, SER531, a characteristic amino acid conserved across species and located within the catalytic center, illustrates a distinctive feature of this particular isozyme.

AbdElmoniem, N. et al. [142] performed pharmacophore modeling and screening of phytochemical compounds from the ZINC database in order to find molecules that simultaneously and efficiently inhibit both HDAC6 and the HSP90 chaperone. Their study led them to three compounds (Figure 21), which were further investigated by molecular docking (Figure 22) to HDAC6 (PDB ID: 6PYE) through Schrödinger’s Glide module. Additionally, molecular dynamics simulations were performed to explore the stability of the complexes and the binding interactions and ADMET analysis, revealing promising results in terms of stability and pharmacokinetic properties. Compound A interacts with the zinc ion via its carbonyl next to the amino group; it also exhibits hydrogen bonding with the amino acid HIE614 and π-cation interactions with the amino acids HIE614, HIE463, and PHE583. It displays a hydrophilic interaction with HIE614 via a water bridge and multiple hydrophobic interactions with the following amino acids: CYS584, TYR745, PHE583, PRO571, PRO464, LEU712, and PHE643. Compound B consists of a benzamide group and interacts with the amino acids GLY582 and TYR745 through hydrogen bonding. At the same time, it develops π-π interactions with PHE643, PHE583, and HIE614, while this compound also shows a variety of hydrophobic interactions with the following amino acids: PHE643, TYR745, PRO571, PHE583, CYS584, PRO464, and LEU712. Compound C similarly features a benzamide group that establishes hydrogen bonds with HIE614, LEU712, and GLY582. Its carbonyl group exhibits affinity for the zinc ion and partakes in π-π interactions with PHE583. Additionally, a water bridge facilitates interactions with HIE614 and LEU712. This compound engages in multiple hydrophobic interactions involving ALA641, PRO464, CYS584, TYR745, PHE643, PHE583, PRO571, PHE642, and LEU712.

### 2.5. HDAC 8

HDAC8, a specific isozyme of the HDAC enzyme, has been proven to play a significant role in the pathogenesis of several life-threatening diseases. These diseases encompass a wide range, such as childhood neuroblastoma, colon cancer, schistosomiasis, Cornelia de Lange syndrome, and various neurodegenerative disorders. The involvement of HDAC8 in these conditions highlights its potential as a therapeutic target for the development of selective inhibitors [143,144,145,146,147]. Remarkably, it has been observed that the approved inhibitor molecules for HDACs have a lesser impact on isozyme 8 compared to other isozymes. This can be attributed to the unique structural characteristics of HDAC8, specifically its first loop (L1), which appears to be shorter compared to other isozymes. This distinct feature allows the loop to move away from the active site, providing the enzyme with increased elasticity and adaptability. Consequently, this structural flexibility creates additional space within the active site of the enzyme, requiring the use of bulkier compounds for effective inhibition [146,148,149,150,151].

The catalytic site of HDAC8 is crucial for its activity, as it contains the zinc ion and facilitates the removal of acetyl groups. This region also interacts with various functional groups, such as hydroxamic acids, which act as inhibitors of the enzyme. The amino acids ASP178, HIS180, and ASP267 are particularly important within this catalytic region, playing essential roles in the enzymatic mechanism [63,152,153]. The presence of a 14 Å tunnel in HDAC8 is a characteristic feature of this enzyme family, and it plays a crucial role in the removal of acetyl groups. This tunnel also serves as an alternative binding site regarding this isozyme [63,81,154]. Located at the terminus of the mentioned tunnel, there is a hydrophobic region that facilitates the release of the acetyl group and forms the third binding site in this isozyme, known as the hydrophobic active site channel (HASC) [155]. The adjacent catalytic site pocket (ACSP) emerges as the fourth binding site, offering an alternative entrance to the enzyme cavity that approaches both the catalytic center and the characteristic tunnel [151].

Sixto-López, Y. et al. [156] performed molecular docking studies on four compounds with several structural differences in the HDAC8 protein after selecting the most suitable structure PDB ID: 3F07 between 20 PDB IDs from the Protein Data Bank according to certain criteria, e.g., the least possible missing amino acids and the minimum amount of mutated residues. The selected protein was prepared, and MD simulations were used to determine its structural changes. Prior to the study, the 3D model was validated using a co-crystallized ligand-APHA (3-(1-methyl-4-phenylacetyl-*1H*-2-pyrrolyl)-N-hydroxy-2-propenamide), yielding an RMSD of 2.485 Å. Docking studies were performed using AutoDock Vina. The compounds investigated in the study included vorinostat (Figure 1), APHA (3-(1-methyl-4-phenylacetyl-*1H*-2-pyrrolyl)-N-hydroxy-2-propenamide), VPA (valproic acid), and tubacin (Figure 23). Notably, tubacin demonstrated the ability to access the catalytic site (CS-HIS180 and HIS143), the adjacent catalytic site pocket (ACSP-PHE152 and TYR306), and the 14 Å tunnel. All of the interactions described above are predominantly π-π interactions. The most prevalent interactions observed for tubacin involved π-π interactions with amino acids TYR306, PHE208, and PHE152. Interestingly, the linker of tubacin exhibited similar behavior to hydroxamic acids, while the cap group of the compound interacted with specific protein surfaces, enabling its entry into the catalytic site during certain simulation periods. Vorinostat interacts exclusively with the catalytic site of the enzyme through π-π interactions, hydrogen bonds, and hydrophobic bonds. Its limited conformational flexibility and the presence of multiple hydrophobic interactions result in a specific orientation within the catalytic site. Key amino acids involved in these interactions include PHE207, PHE208, TYR306, and PHE152. Notably, vorinostat demonstrates a strong interaction with amino acid PHE208, attributed to its phenyl ring. APHA is particularly intriguing as it contains a hydroxamate residue that does not directly interact with the Zn^2+^ ion. Instead, a combination of π-π interactions (with PHE152), hydrophobic interactions (involving LEU179, GLY151, and HIS143), electrostatic interactions (including HIS180, ASP267, and TYR306), and hydrogen bonds (with ASN307) collectively facilitate the compound’s entry into both the catalytic site (CS) and the adjacent catalytic site pocket (ACSP) through the 14 Å tunnel. These interactions occur at different stages of the simulation as the compound navigates its way into the catalytic center through the tunnel. VPA displays the ability to interact with both ACSP and CS. As it enters the catalytic center, VPA establishes strong interactions not only with the Zn^2+^ ion but also with multiple amino acid residues, including ARG37, ASP101, TYR111, TRP141, GLY151, PHE152, TYR154, LEU179, HIS180, PHE208, ASP267, ASP272, SER276, TYR306, and ASN307.

Zhou, H. et al. [80] conducted a study on urushiol (Figure 24) derivatives, which consist of an o-dihydroxybenzene (catechol) and an unsaturated alkyl chain containing 15 to 17 carbons [157]. They observed the structural similarity between urushiol and HDAC inhibitors, e.g., vorinostat, and, considering the known anti-cancer activity of urushiol, they designed three series of urushiol derivatives by: (i) adding the missing zinc binding group-hydroxamic acid group into the tail of the alkyl chain; (ii) introducing hydroxy, carbonyl, amino and methyl ether groups into the alkyl chain; and (iii) replacing the phenolic hydroxy groups by ether or ester groups substituents with different electronic or steric properties into the benzene ring and alkyl side chain. The 30 designed derivatives were subjected to molecular docking studies with the GLIDE program (version 10.2, Schrodinger, LLC, New York, NY, USA, 2015) using HDAC8 (PDB ID: 3SFF) and categorized based on the grid-based scoring function to active and inactive derivatives revealing 10 top scoring compounds (Figure 25) which were further chosen for docking simulation to study their binding pattern. The results revealed that most derivatives (Figure 25) exhibited coordination with the zinc ion through their hydroxyl group, along with moderate hydrogen bonding that contributed to their stabilization. Interestingly, derivatives with substituted electron donors in the alkyl carbon chain showed stronger hydrogen bonding. The researchers also found that the carboxyl and hydroxyl groups of the hydroxamate moiety in the compounds formed hydrogen bonds with GLY140 and HIS143, in addition to their interaction with the zinc ion. Furthermore, the methylene acetal group in some compounds formed hydrogen bonds with PHE152, the nitrogen of the hydroxyl group in Compound **7** formed hydrogen bonds with GLY151, and the addition of a hydroxyl group to the benzoyl ring resulted in hydrogen bonding with TYR154. Furthermore, Compounds 5, 6, and 8 exhibited interactions with the zinc ion through their substituent groups on the aliphatic chain of the linker. In these compounds, the methylene acetal group formed hydrogen bonds with LYS33 and ALA32. Moreover, Compounds **5** and **6** showed additional hydrogen bonding with HIS142 due to the presence of an extra hydroxyl or carbonyl group. Compound **8** displayed additional π-π interactions with TYR154. Compound **21**, which contains an extra amide group, demonstrated binding to the active site of the enzyme through coordination with zinc, as well as hydrogen bonding with GLY151 (*via* the amide group) and TYR306 (via the hydroxyl group). Notably, Compounds **5**, **6**, **7**, and **8** exhibited stronger binding to the HDAC8 enzyme, potentially attributed to the more stable coordination of the oxygen substituents of the linker with the zinc ion.

Finally, the flexibility and the stability of the complexes of HDAC8 with these compounds (Figure 25) was explored through molecular dynamics simulation presenting low RMSD values < 2 Å, verifying the stability and rationality of the initial docking structures. Moreover the flexibility of individual residues was explored through root mean squared fluctuation (RMSF) of backbone atoms for all complexes. These obtained results will guide the synthesis and evaluation of urushiol derivatives as anticancer agents.

Carboxylic acids, although not highly active against HDACs, are recognized for their ability to bind to the metal ion in the enzyme. While carboxylic acids themselves may not be as extensively studied as HDAC inhibitors, certain analog compounds such as valproic acid and butyric acid have been identified as inhibitors of HDACs [152,158,159]. Bora–Tatar, G. et al. [160] examined the activity of carboxylic acids as HDAC inhibitors (HDACIs) and performed molecular docking on the compounds with the highest inhibitory activity, including chlorogenic acid, curcumin, and caffeic acid (Figure 26), specifically targeting HDAC8. AutoDock 4.01 [161,162] was used for the docking studies with HDAC8 (PDB ID: 1T64). In the case of chlorogenic acid, when bound to the enzyme, the phenyl ring of the compound is oriented towards the entrance of the enzyme’s inner cavity. The hydroxyl group of the cyclohexyl moiety forms a hydrogen bond with the amino acid TYR100, while another hydrogen bond is established with PRO35 through the phenolic oxygen. Although chlorogenic acid does not form an ionic bond with the metal ion, its binding energy is highly favorable. Similarly, curcumin also does not interact with the zinc ion, but it forms hydrogen bonds with ASP29 via its hydroxyl group and with TYR100 through the phenolic oxygen. These hydrogen bonds significantly contribute to the favorable binding energy of curcumin. Additionally, curcumin exhibits milder interactions with other amino acids such as ARG37, PRO35, ILE34, and PHE152. In the case of caffeic acid, it is oriented towards the zinc metal at the bottom of the enzyme cavity. It exhibits interactions with TYR306 and PHE152, but the strongest interaction is a hydrogen bond formed through its hydroxyl group with ASP29. Based on this study, it can be concluded that compounds like carboxylic acids do not achieve a strong binding energy primarily through ionic interactions with the metal. Instead, strong hydrogen bonds play a more significant role in their binding affinity to the enzyme.

Kashyap K. and Kakkar, R. [163] conducted molecular docking studies on a series of compounds containing a hydroxamate group and a triazole, which had previously been synthesized by other researchers and showed promising selectivity towards HDAC8. Different crystal structures of HDAC8 were explored with Glide from Schrödinger, Inc., and were validated through the SP methodology, presenting a RMSD values < 2Å. In all compounds and various binding poses, interactions with the zinc ion were observed, along with hydrogen bonds formed with amino acids HIS142, HIS143, and TYR306. The presence of an indole ring as the R1 substituent facilitated additional π-π stacking interactions with the amino acids PHE152 and PHE208. On the other hand, the inclusion of a phenylethyl R2 substitution led to the development of extra π-π stacking interactions with TRP141 and TYR111 through the benzoyl ring, while the presence of a phenyl group at the R1 position resulted in a π-π interaction with PHE208.

As mentioned earlier, Stoddard, S.V. et al. [98] conducted molecular binding studies on various compounds resulting from structural modifications of panobinostat (Figure 8) for HDAC2 and HDAC8. They wanted to explore the molecular characteristics for selectivity against HDAC8, being important for the synthesis of potential biologically active compounds. The Surflex–Dock Geom (SFXC) protocol was applied for the docking, while HDAC8 crystal structure (PDB ID: 1W22) was selected. In this case, the results revealed that the hydroxamate group of panobinostat coordinated with the zinc ion and formed hydrogen bonds with TYR306. Additionally, hydrogen bonding occurred with the amino acid ASP101 through the nitrogen of the compound’s linker group. The compound also engaged in two π-π interactions with the amino acids PHE152 (face-to-face π-π stacking interaction) and PHE208 (parallel π-π interaction). When the central ring of the compound was replaced by a pyrazine ring (TOI1 compound), the binding mode to the enzyme remained quite similar to that of Panobinostat. Once again, in this isozyme, the pyrazine ring was flatter compared to Panobinostat, allowing for a closer approach to PHE208 through π-π interactions. When the central ring of the compound was replaced by a purine ring (ETS5 compound), the hydroxamate group formed a hydrogen bond with HIS143, while the purine ring developed π-π interactions with PHE152. Additionally, the indole ring formed a hydrogen bond with the backbone of GLY151 through a molecular water bridge, and the amine of the linker group contributed to a hydrogen bond with ASP101. The quinoxaline ring compound ETS4 exhibited a conformation that allowed it to interact with the other PHEs, similar to the interactions observed with other compounds. It also showed an additional interaction with HIS180. Regarding the compounds with an imidazole ring (ETS1) and an oxazole ring (ETS2), they also demonstrated specific binding to isozyme 8 of the enzyme. Compound ETS1 oriented its indole ring to the right side of the enzyme entrance through a hydrogen bond with GLY184. However, the imidazole ring exhibited less-favorable π-π interactions (T-shaped) with amino acids PHE152 and PHE208. ETS2 exhibits a similar orientation to ETS1, and its oxazole ring also forms identical T-shaped π-π bond interactions with PHE208. Similarly, the compounds with isoxazole (ETS3) and pyrazole ring (TOI4) also demonstrate T-shaped π-π interactions with the two PHEs, namely PHE208 and PHE152 in the backbone of the enzyme’s active site. Their hydroxamate group forms hydrogen bonds with TYR306, HIS143, and HIS142. Additionally, for both molecules the amine linker forms hydrogen bonds with ASP101, and the indole ring forms a hydrogen bond with PHE152. The study concludes that the six-membered rings in ETS5 and ETS4 are more suitable for binding to isozyme 8, likely due to the wider groove in the enzyme’s formation for this specific isozyme.

Trivedi, P. et al. [164] conducted a research study indicating that more bulky and hydrophobic inhibitor compounds exhibit greater activity against HDAC8. In response, they designed, synthesized, and biologically evaluated derivatives of vorinostat, wherein the linker group was modified to include a piperidine or piperazine ring, and alkyl side chains were introduced as substituents. The three compounds, namely A, B, and C (Figure 27), which exhibited significant activity against the specific isozymes, were subjected to molecular docking analysis to HDAC8 (PDB: 1T69) and using SSchrödinge’s Glide, Inc. Their chemical structures and the docking results, illustrated in the figure once again demonstrated that bulky compounds can effectively fit into the HDAC8 enzyme but encounter difficulties entering HDAC3. In compounds B and C, the hydroxyl group of the hydroxamate moiety established a coordination with the zinc ion through a salt bridge. Additionally, the quinidine and biphenyl groups of Compounds A and B, respectively, engaged in π-stacking interactions with the amino acid TYR306 of the β-chain. Moreover, the terminal phenyl group of the biphenyl segment in Compound B formed another π-p stacking interaction with PHE152 of the same chain. Furthermore, the amino acid TYR306 of the A chain was involved in hydrogen bonding with the carbonyl groups of B and C, as well as with the amide of the hydroxyl group of A. The carbonyl group of A was positively charged and participated in a p-cationic interaction with HIS180. Additionally, the hydroxyl group of A acted as a hydrogen donor for the amino acid HIS143. Notably, the charged amide group of piperidine in Compound A engaged in a p-cationic interaction with PHE208. In addition to these interactions, the influence of water molecules appeared to be significant in determining the activity of these molecules.

Giannini, G. and his research team [165] set out to investigate HDAC enzyme inhibitors containing a thiol group, specifically a protected thiol group, as opposed to a hydroxamate residue. It is important to note that thiol functionality is rare among HDAC inhibitors, with romidepsin (Figure 2) being a notable exception, in the form of a prodrug disulfide. The limited research on thiol groups, coupled with the challenges posed by their vulnerability to oxidation and reactivity, made them an intriguing subject for investigation. In addition, this investigation was motivated by the fact that hydroxamate derivatives are classified as being primarily effective in the treatment of hematological cancers, with more limited efficacy against solid tumors [166].

The primary aim of this research was to synthesize novel derivatives designed for HDAC inhibition. Particular attention was paid to the substitution of the methylene linkage of the inhibitors at the ω-position, with the aim of potentially increasing their activity and selectivity. In this context, the study investigated the use of lactam carboxamides [167] as substituents at the aforementioned position. At the same time, the role of the cap group in these compounds was investigated by considering alternative groups such as benzyl, phenethyl and cyclopentyl rings as replacements for the conventional phenyl ring.

The results of the biological tests indicate that compounds with a 6-carbon linker exhibit superior antiproliferative activity. Surprisingly, the lactam group did not appear to play a significant role in this activity. Instead, the stereochemistry of the ω-position emerged as a dominant factor in HDAC enzyme inhibition, with an essential requirement for an S-configuration, likely to mimic the natural peptide substrate of the enzymes.

However, it is important to note that isozyme 8 showed a peculiar behavior. In this case, the cap group, and in particular derivatives such as trifluoromethylbenzyl, ben-zyloxybenzyl, and m-methylphenylethyl, significantly influenced the activity of the compound. These compounds (Figure 28) displayed selectivity against isozyme 8, whereas compounds with excellent activity against other isozymes showed limited efficacy here. Molecular docking studies were carried out to gain insight into the interaction of compound D (Figure 28) with the enzyme. It was observed that the benzyloxy-benzyl substituent of the compound could access a unique pocket of the enzyme which is located close to the L1 loop (near amino acid LYS33). Importantly, this feature is specific to HDAC8 and is not shared by other isozymes. In addition, HDAC8 is known to have a shorter L1 loop than other isozymes, resulting in a wider active site with greater surface area and flexibility [151]. In addition, the lactam ring and nitrogen atom interacted with ASP101 and the benzyl moiety of the molecule approached a secondary pocket of the enzyme. It was observed that the benzyloxy-benzyl group was exposed to the solvent when the same molecule was tested for interaction with HDAC3.

The Schrödinger software suite was used for molecular modelling studies. Ligands were generated using Maestro 9.6 [168] and Ligprep 2.8 [169], while protein structures were refined using the Protein Preparation Wizard [170]. Docking studies were performed with Glide 6.1 [171,172] using the SP scoring function, with default settings unless specified. The docking grids were centered on co-crystallised molecules with HDAC3 and HDAC8. Ligands were prepared using Ligprep 2.8 [169] followed by energy minimization using the OPLS2005 force field with Macromodel 10.2 [168] The best pose for each ligand was retained. Molecular dynamics (MD) simulations were performed on the top-ranked complexes from the docking.

Tilekar, K. et al. [173] undertook the synthesis of a series of thiazolidinediones compounds strategically designed for the dual inhibition of PPARs (peroxisome proliferator-activated receptors) and HDACs. Thiazolidinediones are widely recognized as a class of PPARγ antagonists. In a prior study, these investigators successfully synthesized (*Z*)-2-((1-((2,4-dioxothiazolidin-5-ylidene)methyl)naphthalen-2-yl)oxy)-N-(4-nitrophenyl)acetamide (Compound **I**, Figure 29) featuring a naphthylidene linker and thiazolidinedione group, which exhibited robust inhibitory activity against HDACs. Specifically, this compound demonstrated an IC_50_ value of 2.3 μM for HDAC8 [174]. Thus, they retained the naphthylidene linker and synthesized compounds with different cap groups, such as heterocyclic, heteroaryl, or extended aromatic moieties, to develop selective inhibitors. The biological activity against HDAC4 and HDAC8 was evaluated. Most of the compounds exhibited significantly enhanced activity against HDAC4 and much more moderate activity against the other enzyme. The most potent compound against HDAC8 was Compound B (Figure 29).

The probable reason for this result is the change in the substitution position of the naphthylidene group, specifically where the thiazolidinedione group was introduced. This change altered the *L*-configuration of the molecules, resulting in loss of activity against isoenzyme 8 [175]. In addition, placing the thiazolidinedione group more distant from the naphthylidene structure caused the compounds to adopt a more extended conformation. To further investigate this phenomenon, a docking study was carried out using the HDAC8 crystal structure (PDB ID: 3SFF) previously used to study Compound **I** [174]. The docking analysis showed a remarkable overlap with an RMSD (root mean square deviation) of only 0.2 Å over 26 heavy atoms. The results of the docking study revealed comparable interactions of these compounds with HDAC8 to those observed for Compound **I**. In particular, the carbonyl oxygen of the molecules interacts with the zinc ion within the catalytic site. The naphthalene group forms π-stacking interactions with PHE152 and PHE208 and a π-sulphur interaction with MET274. In contrast, the thiazolidinedione group is exposed to the solvent. The remarkable structural flexibility of the HDAC8 enzyme allows for the accommodation of more branched molecules, as exemplified by Compound **I** [176,177]. It is worth noting that the docking study was carried out using MOE 2019 software, while modifications to the ligands and protein were performed via Amber14 force field calculations. The interactions were then analyzed using the Computed Atlas of Surface Topography of Proteins (CASTp).

In response to the inability of hydroxamate derivatives to bind selectively to a specific metal and their unfavorable binding to other targets, Muthyala, R. et al. [178] decided to investigate another chemical group, namely 1-hydroxypyridine-2-thiones (1HPTs), which exhibit selectivity coordinating with zinc. In their previous research, they had already synthesized 1HPT analogs containing amino acids that inhibited a zinc dipeptidase associated with antibiotic resistance [179]. Hence, these synthesized a series of compounds which were biologically evaluated to determine their selective activity against specific HDACs isozymes. Compounds **1** and **2** (Figure 30) showed effective inhibition and high selectivity against HDAC8. To gain further insight, molecular docking studies were performed using the Schrödinger modelling package, the crystallographic structure of HDAC8 (PDB ID:1T69), and the OPLS 2005 force field. The compounds were first deprotonated, and then molecular dynamics were used to optimize their octahedral geometry. An explicit TIP3 water-box was used for these simulations.

As demonstrated, the binding molecules were adapted internally to fit the hydrophobic channel of the enzyme. This, along with the maintenance of the octahedral configuration, played a critical role. In (1-hydroxy-6-thioxo-1,6-dihydropyridine-2-carbonyl)-D-alanine (Compound **1**), the carboxyl ion formed a hydrogen bond with HIS180, which was coordinated to zinc. However, this interaction did not occur in (*R*)-2-(1-hydroxy-6-thioxo-1,6-dihydropyridine-2-carboxamido)-2-phenylacetic acid (Compound **2**) due to the bulkier phenyl group. The rotatability around the amide bond was also crucial for the development of bonds with PHE207 and PHE208, as well as the formation of a salt bridge with LYS33 through the carboxyl group.

Galletti, P. et al. [180] carried out the synthesis of azetidinones with a focus on taking advantage of the properties of β-lactams and a thorough review of previous studies demonstrating their efficacy as HDAC inhibitors [181]. After biological evaluation, it was found that azetidinones bearing the *N*-thiomethyl group showed superior inhibition of the HDAC8 isoenzyme, while enantiomerism also had a major role to play.

A computational analysis was carried out, starting with the calculation of the geometry and energy of the zinc ion in complex with the ligand using the PDB ID: 1T67 as the crystallographic protein form and compound A (Figure 31) in the *R* stereoisomeric configuration as the focal ligand. A molecular dynamics calculation was then performed for the complex comprising the ligand, a water molecule, and the protein. In particular, this analysis showed that the ligand maximized its interactions with the enzyme, with its cap group forming hydrogen bonds with PHE207 and its carbonyl group interacting with the zinc ion. Furthermore, calculation of the binding energy of the complex provided additional support for the selectivity of these compounds towards HDAC8. In addition, a study of pairs of enantiomers, together with the calculation of free and binding energies for their complexes, revealed that in one enantiomer (in which the side chain is oriented downwards), the side chain interacts with TRP141 via sulfur, whereas in the other enantiomer, it does so via a methyl group. An ab initio Hartree–Fock 3-21G single point calculation was used for the geometric aspects of the analysis, and Glide software (version 4.0 of Schrödinger) was used for the docking procedure using the SP protocol.

Ononye, S. N. et al. [182] synthesized tropolone analogues to evaluate their potential as HDAC inhibitors and assess their selectivity. This structural group was chosen because of its improved metabolic stability compared to rapidly reduced hydroxamic acids [183,184]. Tropolones are characterized by strong metal associations [185]. Among the synthesized compounds, four demonstrated significant inhibitory activity against HDAC8 (Figure 32). Molecular docking studies using Surflex-Dock software and the crystallographic protein form with PDB ID: 1W22 revealed that when a bulky and hydrophobic substituent was present at the β-position, it was accommodated in the hydrophobic pocket of the enzyme formed by PHE152, TYR306, MET274, and LYS33, with a notable hydrogen bond formation between the carbonyl and TYR306. The enzyme pocket was particularly spacious, and interactions such as those with *β*-substituted cyclopentyl were difficult to form. Increased activity was associated with branched molecules bearing the bulkiest substitutions. Compounds with α-substitutions also entered the hydrophobic pocket involving TYR306, MET274, and PHE152, but greater bulk, as in the case of naphthalene, led to destabilized interactions.

## 3. Conclusions

In summary, HDAC enzymes play a pivotal role in numerous epigenetic regulations, affecting various genes and contributing to the pathogenesis of several diseases, particularly cancer and neurodegenerative disorders. Isozymes 1, 2, 3, 6, and 8 are particularly critical due to their association with the above-mentioned diseases. Their inhibition thus holds therapeutic potential. Ongoing research into the inhibition of this group of enzymes suggests promising avenues for therapeutic approaches, with a focus on selective isozyme inhibition to minimize pleiotropic effects and effectively address a broad range of dysfunctions, an effort that has yet to be fully achieved.

Computational chemistry tools, particularly molecular docking, are extensively used in drug design and development, providing a solution for creating more selective HDAC inhibitors for each isozyme by elucidating the binding mechanism of each chemical compound with its target. As in the case of HDACs, the distinct structural features of individual enzymes or enzyme isozymes become apparent. This process assists in the design and precise control of binding affinity by elucidating the necessary structural features that potential therapeutic compounds should possess in order to enhance their efficacy. These tools simplify the process of screening thousands of compounds for potential activity against therapeutic targets, while also identifying structural attributes that enhance their maximal impact. Essentially, this simplifies the drug design process and enhances the development of novel targeted drugs. Molecular docking studies of the HDAC isozymes have revealed the different characteristics that result from the interaction between the inhibitory molecules and each isozyme. Different interactions of each isozyme with potential inhibitors describe effective inhibition. These interactions involve different amino acids, emphasizing the structural differences between HDAC isoforms.

For the HDAC1 isozyme, which plays a key role in many malignancies, it seems that inhibition depends on the structure of its 14 Å tunnel, so research focuses on the development of selective inhibitors interacting with the catalytic center. In this review, three different docking studies on HDAC1 are included using HDAC1 (PDB ID: 4BKX) applying AutoDock 4.2 [83,85], AutoDock 4.2 and AutoDock4Zn [85] (an improved force field of AutoDock for docking of Zinc Metalloprotein), and Schrödinger (2019–4) Maestro v12.2 [96]. Besides using different docking programs, it can be concluded from all of the docking results that amino acid GLY149 at the binding pocket entrance, together with PHE150 and PHE205, plays a critical role in interaction, since they contribute to the definition of the final shape of the channel. This is consistent when studying different compound classes as already mentioned and using different reference compounds. Other amino acids, such as HIS141, HIS178, ASP264, and TYR303, seem also to contribute to isozyme-specific inhibitor selectivity. Moreover, the biological results seem to be in accordance with the docking results.

Isozyme 2 is implicated in many cancers and Alzheimer’s disease. Four docking studies for HADAC2 are included in this review using (a) different reference compounds, e.g., panobinostat, vorinostat, resveratrol, and entinostat and (b) docking programs such as Surflex-Dock Geom (SFXC), AutoDock 4.2, Schrödinger (2019–4) Maestro v12.2, and MOE. The selected protein from the PDB was the HDAC2 (PDB ID: 4LXZ) [83,98,101] in all of the cases except that from the Silva Urias, B. et al. study using HDAC2 (PDB ID: 4LY1) [96]. HDAC2 is characterized by an internal active site tunnel cavity with PHE155 and PHE210 on two opposite sides. Even when docking different reference compounds and synthesized derivatives with a variety of programs, it seems that selective inhibition is based on the development of π-π interactions with these amino acids. Other characteristic interactions being developed during inhibition are hydrogen bonds with amino acids TYR308, HIS145, and HIS146. In two [91,101] of the four studies included in this review, biological evaluation of HDAC2 is performed, confirming the docking results.

HDAC isozyme 3 stands out for the fact that besides cancer it participates in memory-related processes while structurally having a much smaller binding pocket. Results from different software such as the Glide module of Schrodinger Maestro, GOLD and Glide implemented in Schrödinger Suite [123], and different series of compounds using HDAC3 (PDB ID: 4A69) seem to be in accordance. Its inhibition is stabilized by the formation of a hydrogen bond with the amino acid residue TYR298. Other distinctive interactions appear to be with amino acids PHE144, HIS134, and PRO200. In one of the tree studies mentioned above [119], docking results are supported by biological results revealing the selectivity of several benzamide derivatives containing aryl or heteroaryl groups for HDAC3 isozyme over the other HDAC isozymes.

HDAC isozyme 6 displays the distinctive feature of having two structural domains, while its inhibition contributes to the reversal of Alzheimer’s pathogenesis through tau proteins. In this review, different studies are discussed with different compound classes obtained from libraries, HDAC6 (PDB ID: 5EDU, 5EF7, and 6PYE), and software such as GOLD v5.2.2., Autodock Vina, and Schrödinger’s Glide. Besides these differences, it seems that HDAC6 selective inhibition is characterized by hydrogen bond formation with the amino acids HIS610 and HIS614 but also by the development of more polar interactions. These are stabilized by hydrophobic interactions with amino acids of the catalytic center such as PHE680, LEU749, and TYR782.

HDAC isozyme 8 being involved in the pathogenesis of several life-threatening diseases demonstrates a pronounced conformational flexibility, necessitating the engagement of more bulky compounds for selective inhibition. For docking studies, different HDAC8 (PDB IDs: 3F07, 3SFF, 1T64, 1W22, and 1T69) are used with software such as AutoDock Vina, AutoDock 4.01, Surflex-Dock Geom (SFXC), and GLIDE Schrödinger, Inc. The liability of the docking studies was confirmed with molecular dynamics simulations with RMSD values lower than 2 Å. In seven of the ten research studies, docking simulations support the biological results. HDAC8 is distinguished by interactions mediated by specific amino acids, notably TYR306, PHE208, PHE152, and TYR154.

Finally, it has to be noted that for each isozyme, the interactions critical for enzyme inhibition are enhanced by the interaction or coordination of the molecules with the Zn^2+^ ion located in their active sites. This interaction can be achieved through the carbonyl, amine, or hydroxyl groups.

It is interesting to note that this particular isozyme has a smaller loop L1, which, as mentioned above, results in a wider active site. As a result, this enzyme is more effectively inhibited by bulkier molecules. As already shown, inhibitors for this isoenzyme can be successfully synthesized that do not contain characteristic groups such as hydroxamic acids, but instead sulphur, lactams or tropolones. In each case, the size of the molecule and the presence of structural diversity and branching favor the efficacy of the inhibitor. In addition, the studies have illustrated that the cap group of inhibitors can include different ring structures beyond aromatic rings, including heteroaromatic, heterocyclic, or benzyloxy-benzyl groups.

It is also important to understand that enzyme flexibility can lead to different enzyme conformations, and this aspect needs to be considered. HDAC8 switches between open and closed conformations, and these structural changes significantly affect ligand binding. Including multiple receptor structures in the docking process is important to clarify the exceptional flexibility of HDAC8 and lead to more accurate predictions of ligand interactions. For HDAC8, studies have shown that ligands with flexible surface binding groups tend to prefer the open conformation of the protein. This preference is due to the ability of the open conformation to protect the ligands from solvent exposure during binding. This observation helps to explain why pyrazole-based inhibitors are more potent than their isoxazole derivatives. By using multiple receptor structures, the docking algorithm can explore a range of conformations, ensuring that the most appropriate structure is selected for each specific ligand [186].

In all cases, a comprehensive study of the binding mode of inhibitors to HDACs isozymes is essential for refining their design and improving their selectivity.

## Figures and Tables

**Figure 1 pharmaceuticals-16-01639-f001:**
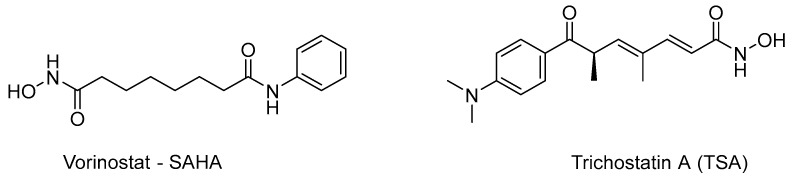
Molecular structures of vorinostat and trichostatin.

**Figure 2 pharmaceuticals-16-01639-f002:**
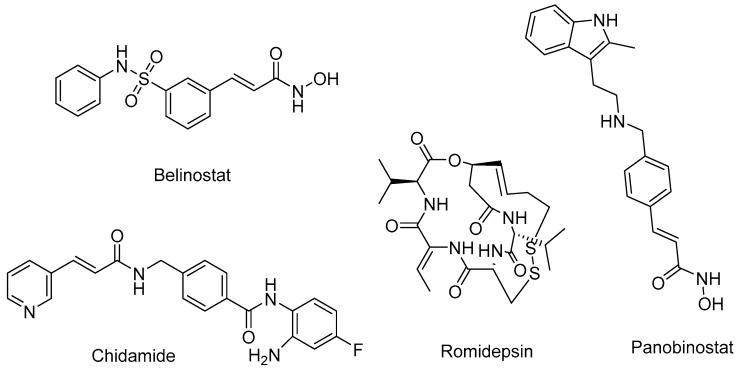
Molecular structures of belinostat, panobinostat, romidepsin, and chidamide.

**Figure 3 pharmaceuticals-16-01639-f003:**
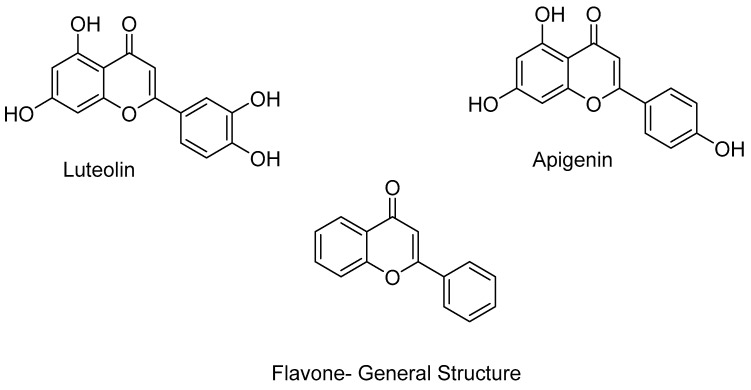
Chemical structures of Luteolin, Apigenin and Flavone.

**Figure 4 pharmaceuticals-16-01639-f004:**
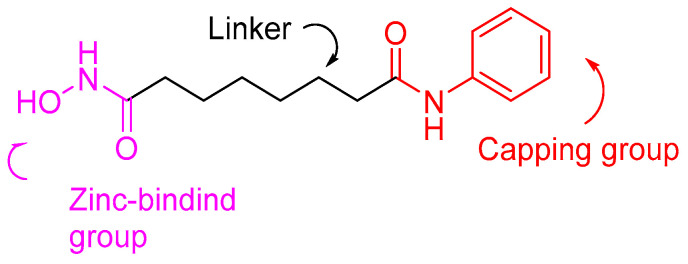
The example of vorinostat clearly demonstrates the distinct presence of the three structural regions of HDAC inhibitors.

**Figure 5 pharmaceuticals-16-01639-f005:**
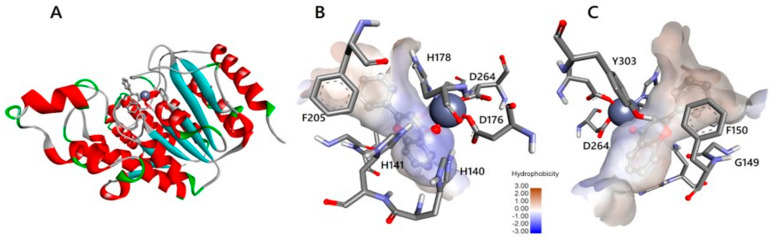
Scafuri et al. conducted molecular docking studies to elucidate the binding mechanisms of flavones to the HDAC1 isozyme. Figure (**A**) depicts the interaction between flavone and HDAC1, with the zinc ion represented as a sphere. Figure (**B**) illustrates the specific interactions between flavone and certain amino acids, with the zinc ion depicted as a sphere interacting with the carbonyl oxygen of the compound. Figure (**C**) presents the previous figure from a different perspective. In the last two figures, the compound has been colorized using a gradient of colors to indicate its hydrophobicity [83].

**Figure 6 pharmaceuticals-16-01639-f006:**
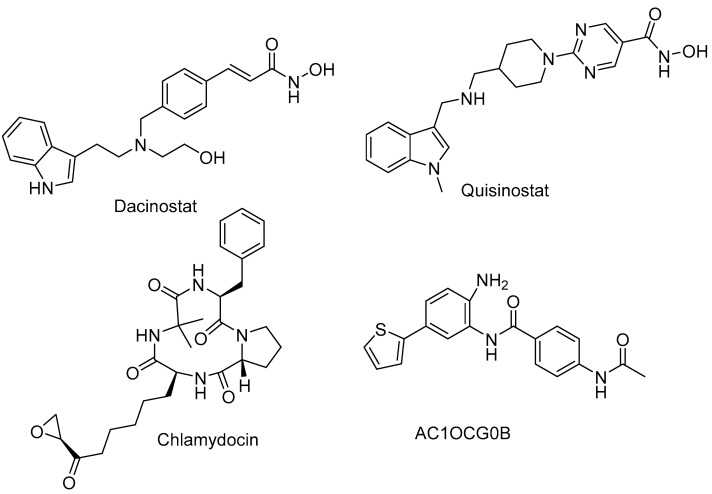
Chemical structures of the compounds whose molecular binding mode to the HDAC1 enzyme was studied by Sixto–López et al. [85].

**Figure 7 pharmaceuticals-16-01639-f007:**
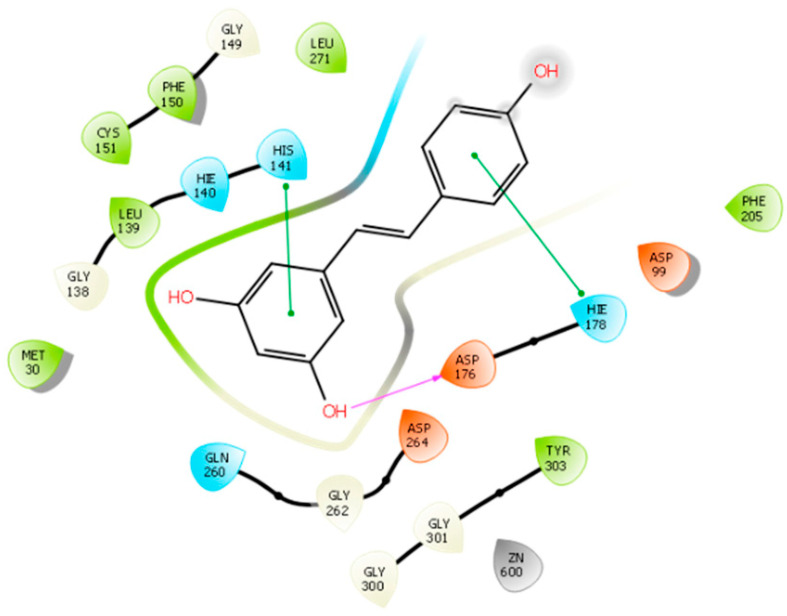
Silva Urias et al. [91] investigated the interactions between resveratrol and its analogues and the HDAC1 enzyme. Hydrogen bond interaction is illustrated by a purple dashed arrow while π-interactions are illustrated with a green line. The hydrophobic residues are shown in dark green, the polar ones (uncharged) in cyan, the negative charged in red, Gly in light gray and zinc in dark gray. The interactions were visualized and presented in a two-dimensional format.

**Figure 8 pharmaceuticals-16-01639-f008:**
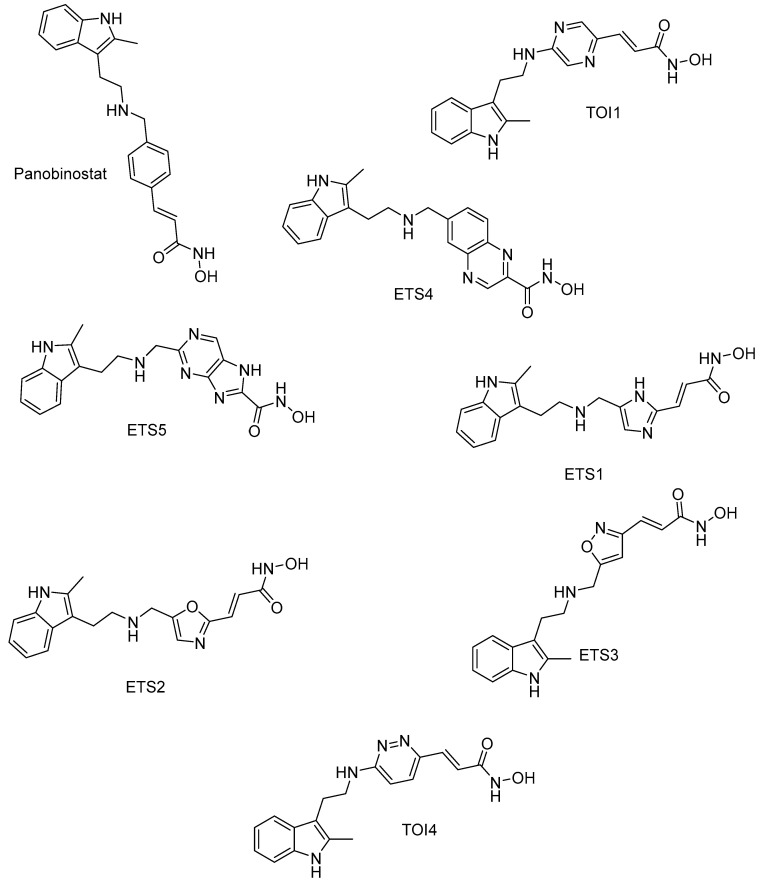
Stoddard et al. synthesized the above compounds (ETS1-5, TOI1, and TOI4) with structural modifications from panobinostat and tested their selectivity towards HDAC isozymes 2 and 8 via molecular docking tools [98].

**Figure 9 pharmaceuticals-16-01639-f009:**
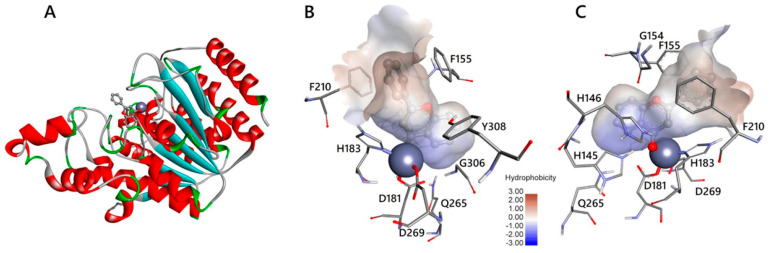
Scafuri et al. conducted molecular docking studies to elucidate the binding mechanisms of flavones to the HDAC2 isozyme. Figure (**A**) depicts the interaction between flavone and HDAC2, with the zinc ion represented as a sphere. Figure (**B**) illustrates the specific interactions between flavone and certain amino acids, with the zinc ion depicted as a sphere interacting with the carbonyl oxygen of the compound. Figure (**C**) presents the previous figure from a different perspective. In the last two figures, the compound has been colorized using a gradient of colors to indicate its hydrophobicity [83].

**Figure 10 pharmaceuticals-16-01639-f010:**
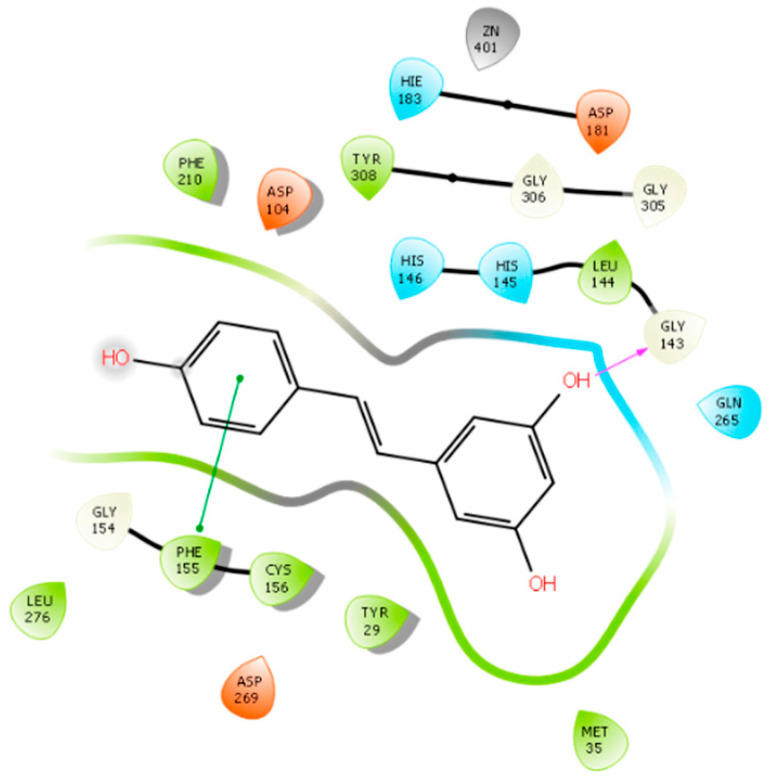
Silva Urias et al. [91] investigated the interactions between resveratrol and its analogues and the HDAC2 enzyme. Hydrogen bond interaction is illustrated by a purple dashed arrow while π-interactions are illustrated with a green line. The hydrophobic residues are shown in dark green, the polar ones (uncharged) in cyan, the negative charged in red and Gly in light gray and zinc in dark gray. The interactions were visualized and presented in a two-dimensional format.

**Figure 11 pharmaceuticals-16-01639-f011:**
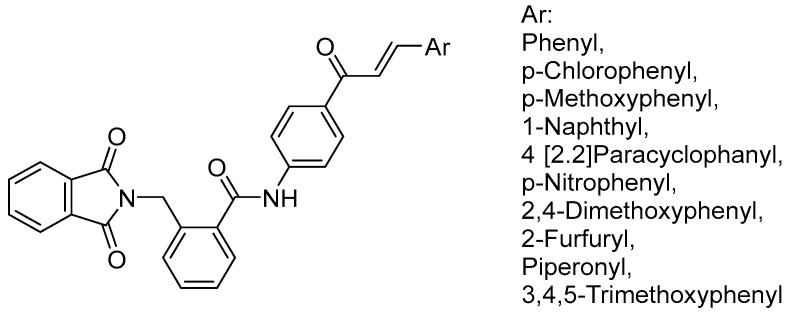
α-phthalimido-substituted chalcones synthesized by Mourad et al. [101].

**Figure 12 pharmaceuticals-16-01639-f012:**
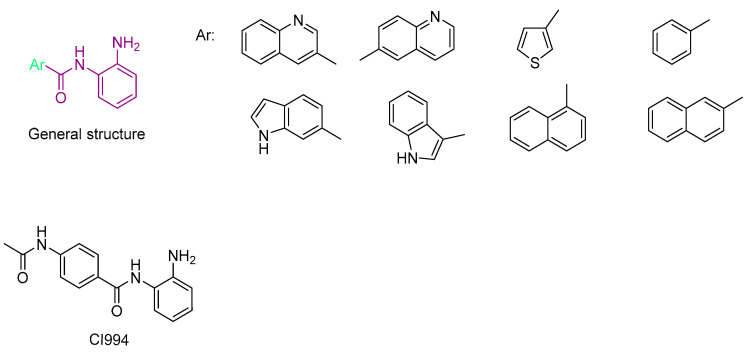
Routholla, G. et al. synthesized and studied compounds with the above general structure considering CI994 as reference compound [119].

**Figure 13 pharmaceuticals-16-01639-f013:**
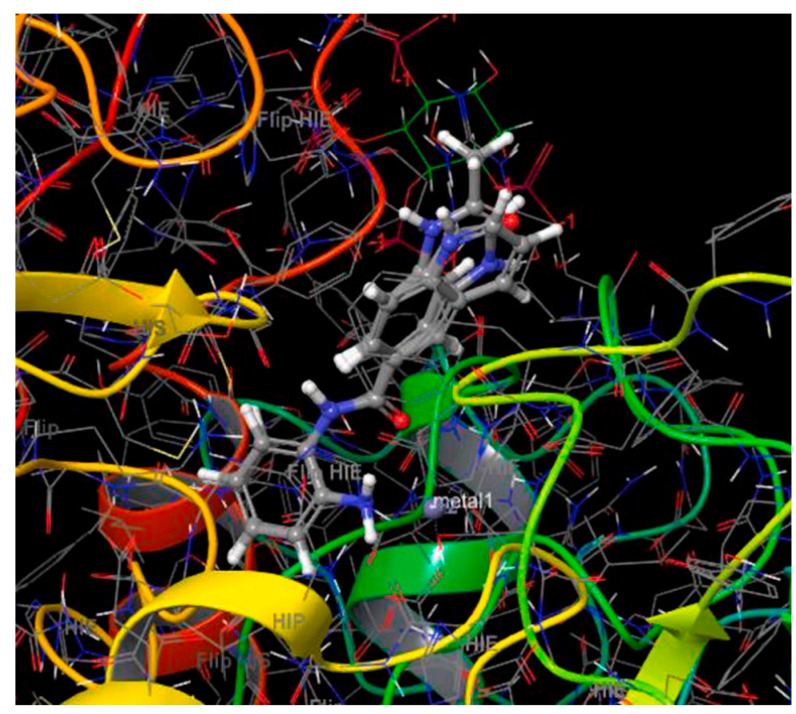
G. Routholla et al. [119] concluded that most of their compounds show almost the same orientation with CI994 when binding to the HDAC3 enzyme, i.e., superposition. Compounds are represented in stick models. The backbone of the compounds is colored in grey (carbon) while oxygen in red, nitrogen in blue and hydrogen in white.

**Figure 14 pharmaceuticals-16-01639-f014:**
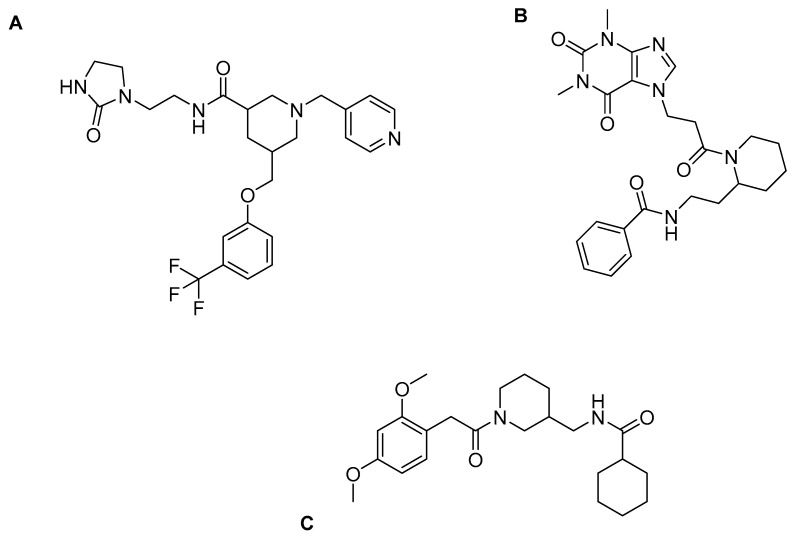
The chemical structures of the hit compounds obtained from the study of Kumbhar et al. (**A**) Hit1, (**B**) Hit2, and (**C**) Hit3 [121].

**Figure 15 pharmaceuticals-16-01639-f015:**
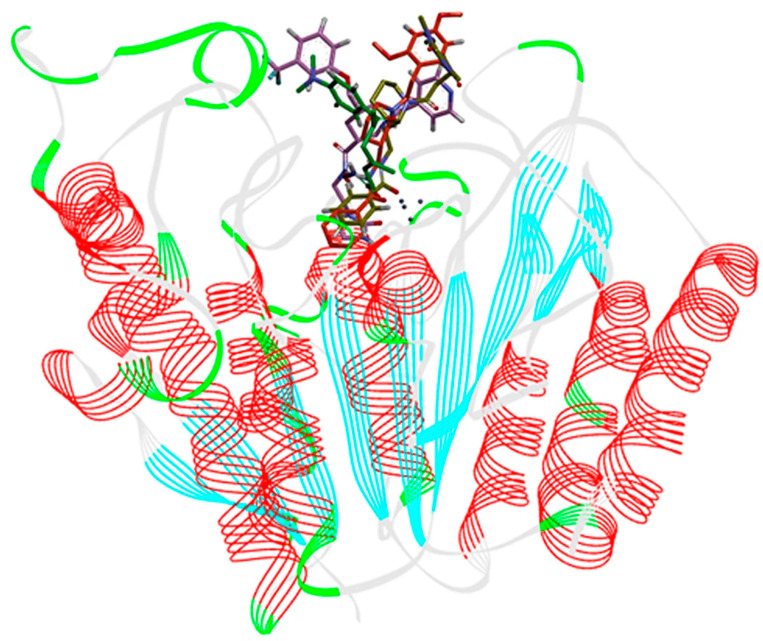
The binding pose of the three compounds and the TSA (reference compound) [121]. Compounds are represented in stick models (Hit1 purple, Hit2 green and Hit3 red).

**Figure 16 pharmaceuticals-16-01639-f016:**
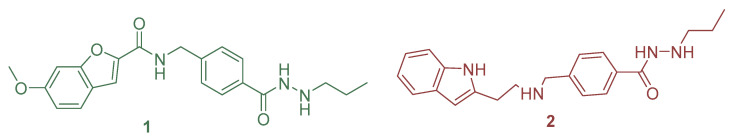
Chemical structures of Compounds **1** and **2**, the mode of binding to the HDAC3 of which was examined by Emre F. Bülbül et al. [122].

**Figure 17 pharmaceuticals-16-01639-f017:**
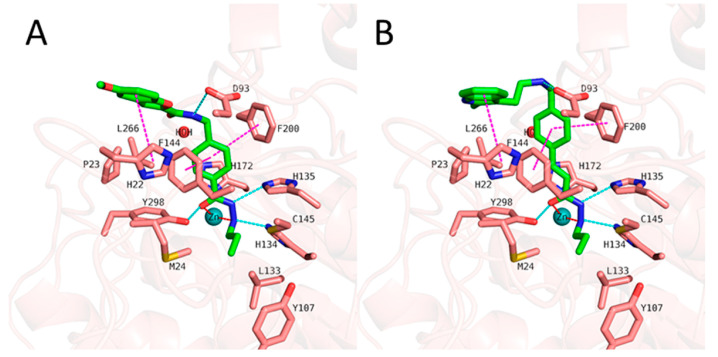
Compounds **1** (**A**) and **2** (**B**) in binding with the enzyme HDAC3. The compounds are represented by green sticks while the amino acid residues of the enzyme with which they interact are represented by coral sticks. Zinc is represented by a cyan sphere [122].

**Figure 18 pharmaceuticals-16-01639-f018:**
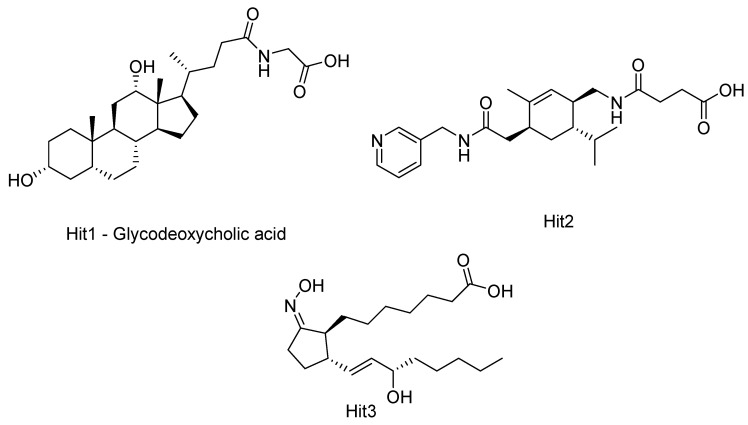
The chemical structures of the hit compounds studied by Amir Zeb et al. [139].

**Figure 19 pharmaceuticals-16-01639-f019:**
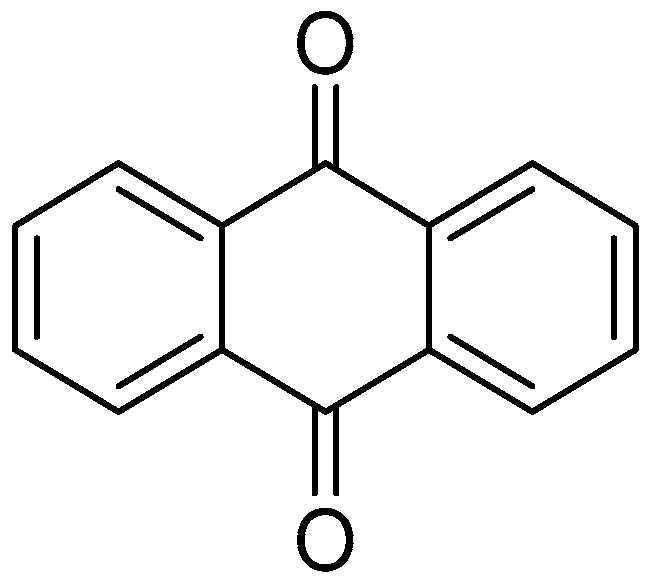
Chemical structure of anthraquinone.

**Figure 20 pharmaceuticals-16-01639-f020:**
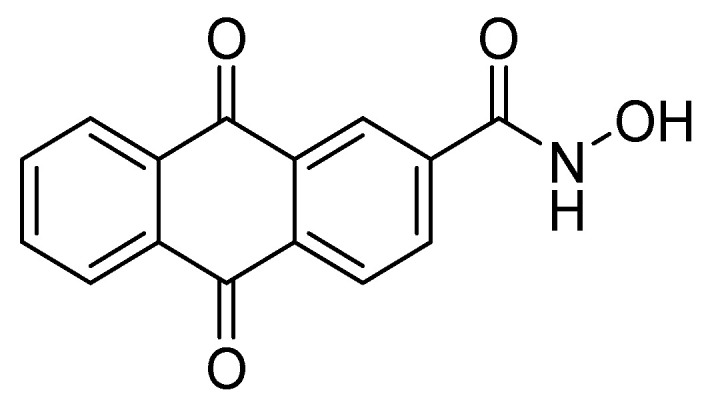
The chemical structure of the compound designed and synthesized by Song, Y. et al. and exhibiting selective inhibition against HDAC6 [140].

**Figure 21 pharmaceuticals-16-01639-f021:**
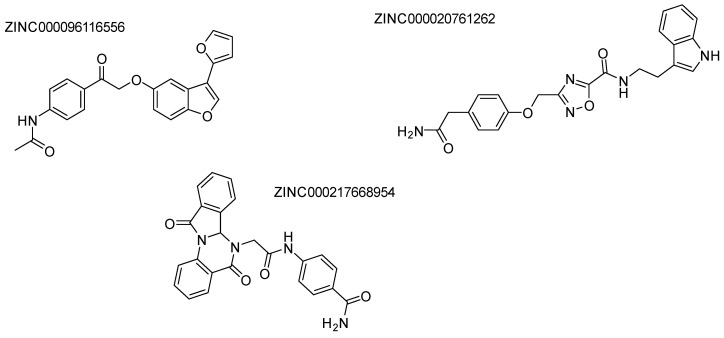
ZINC000096116556, ZINC000020761262, and ZINC000217668954 correspond to compounds A, B, and C taken from N. AbdElmoniem et al. The binding mechanisms of these compounds with HDAC6 and HSP90 enzymes have been extensively investigated [142].

**Figure 22 pharmaceuticals-16-01639-f022:**
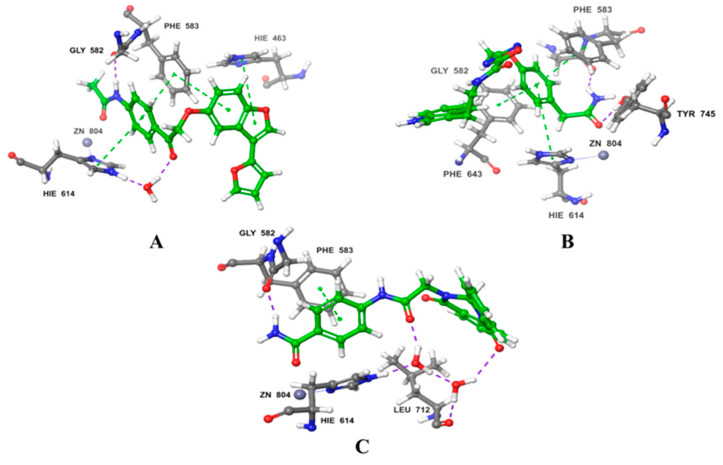
Three-dimensional representation of the interactions of the three compounds A, B, C with the HDAC6 enzyme by the AbdElmoniem, N. et al [142].

**Figure 23 pharmaceuticals-16-01639-f023:**
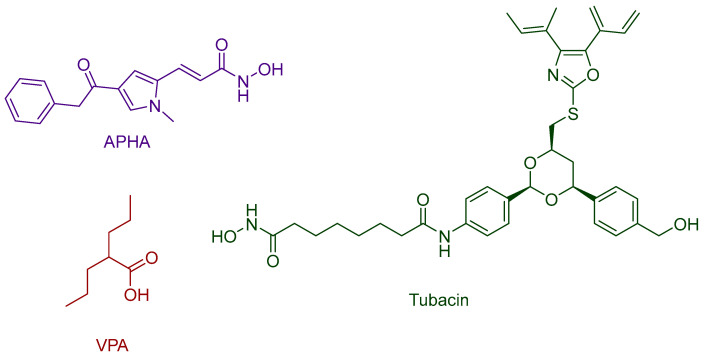
Molecular structures of APHA, VPA, and tubacin.

**Figure 24 pharmaceuticals-16-01639-f024:**
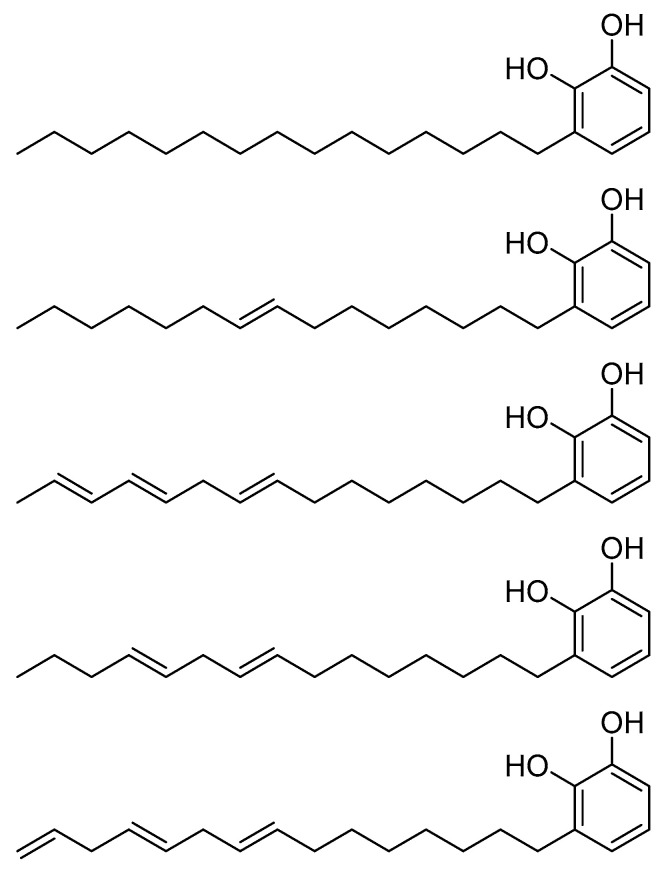
Urushiol general chemical structures.

**Figure 25 pharmaceuticals-16-01639-f025:**
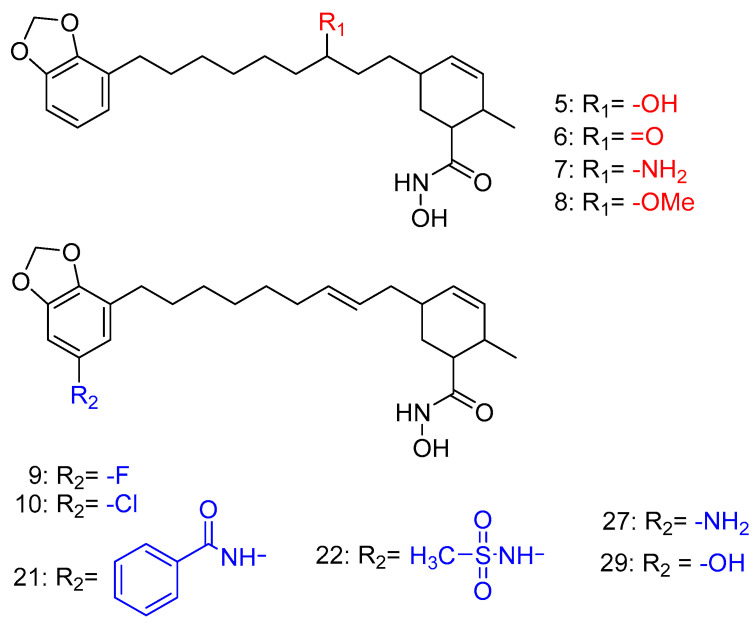
Urushiol derivatives which have been synthesized and on which molecular docking studies have been carried out by Zhou et al. [80].

**Figure 26 pharmaceuticals-16-01639-f026:**
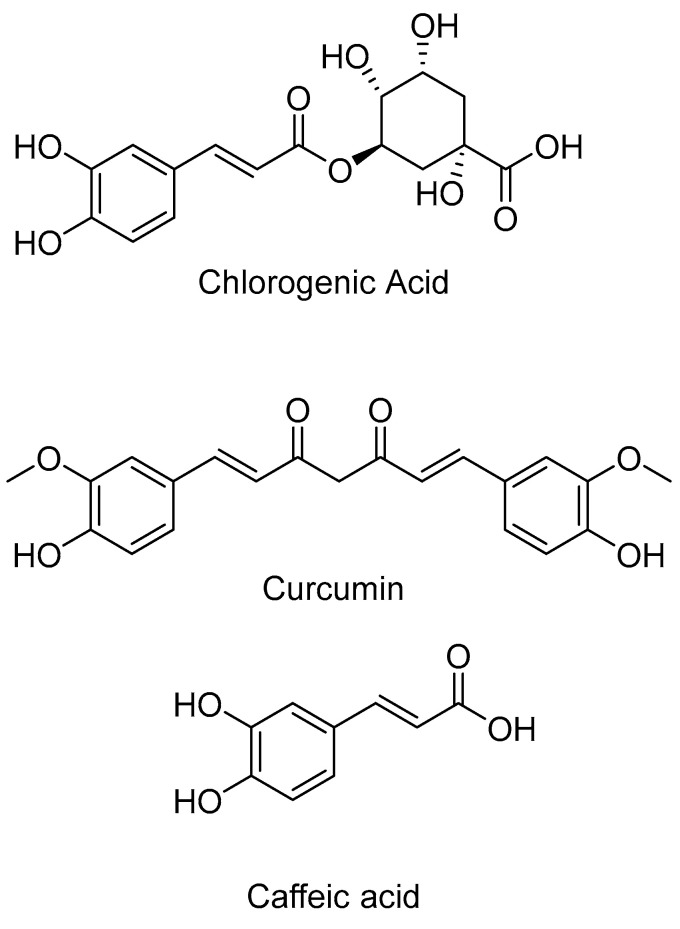
Chemical structures of chlorogenic acid, curcumin, and caffeic acid.

**Figure 27 pharmaceuticals-16-01639-f027:**
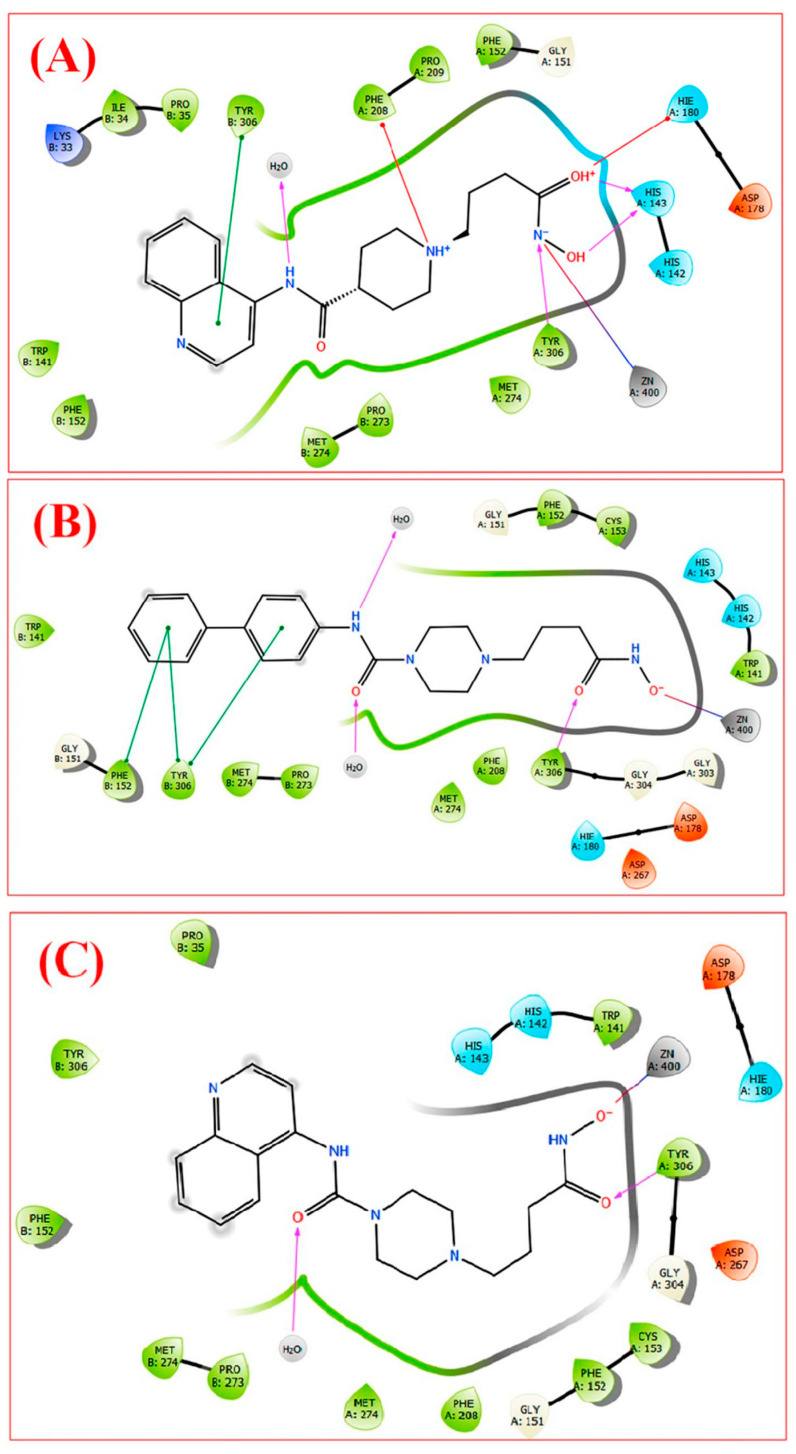
P. Trivedi et al. conducted molecular docking studies on 3 compounds that seemed promising for selective activity on HDAC8. Shown above are the interactions of compounds (**A**–**C**) with the enzyme in two-dimensional form [164]. Reprinted with permission from Copyright {2023}.

**Figure 28 pharmaceuticals-16-01639-f028:**
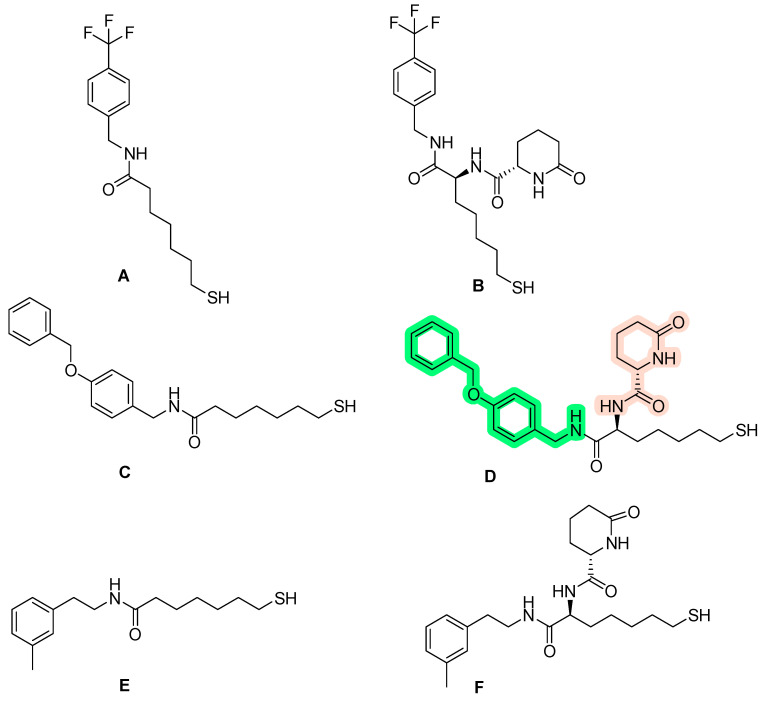
The compounds were synthesised by Giannini, G. et al. [165] and demonstrated selectivity for the HDAC8 isoenzyme. Compounds (**A**,**B**) contain a trifluoromethylbenzyl moiety, compounds (**C**,**D**) contain a benzyloxybenzyl moiety, and compounds (**E**,**F**) contain an m-methylphenylethyl moiety. In particular, compound (**D**) was tested for interaction with the enzyme. Its lactamate moiety is shown in light pink, while the cap group is shown in light green.

**Figure 29 pharmaceuticals-16-01639-f029:**
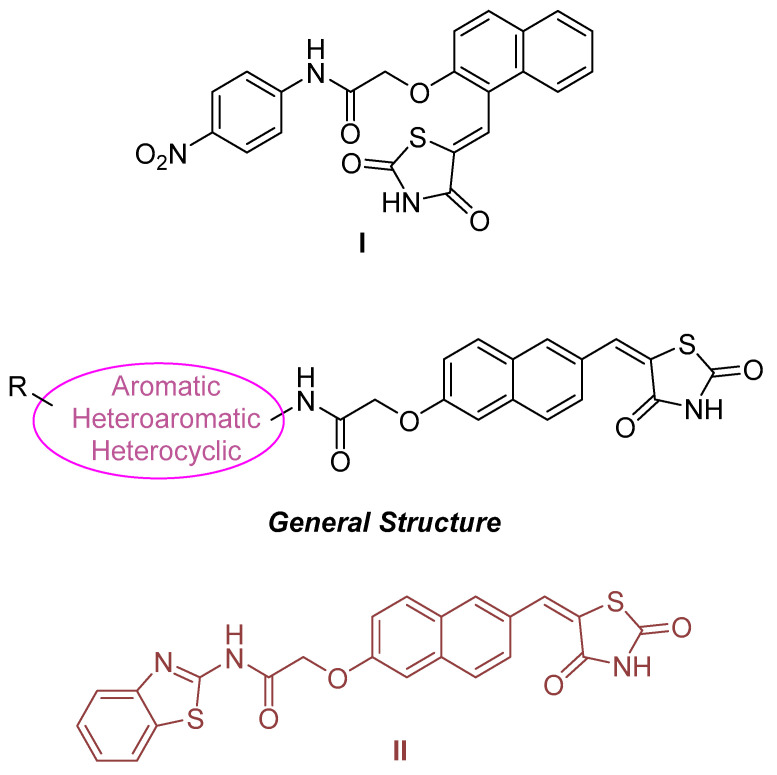
(*Z*)-2-((1-((2,4-dioxothiazolidin-5-ylidene)methyl)naphthalen-2-yl)oxy)-N-(4-nitrophenyl)acetamide (Compound **I**) was previously synthesized by Tilekar, K. et al. [173], and the new series of compounds they generated shared the general structure above. Among these compounds, (*E*)-*N*-(benzo[d]thiazol-2-yl)-2-((6-((2,4-dioxothiazolidin-5-ylidene)methyl)naphthalen-2-yl)oxy)acetamide (Compound **II**) showed the highest activity against HDAC8.

**Figure 30 pharmaceuticals-16-01639-f030:**
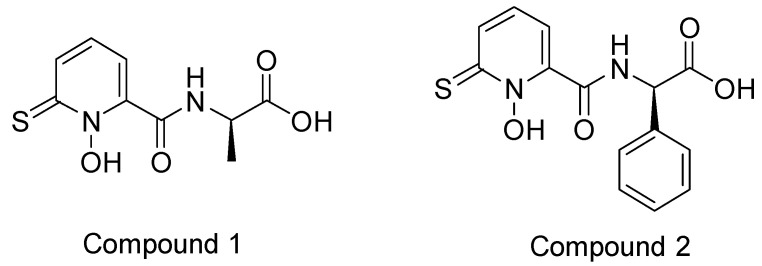
Compounds **1** and **2** synthesized and evaluated for biological activity against HDACs by Muthyala, R. et al. [178].

**Figure 31 pharmaceuticals-16-01639-f031:**
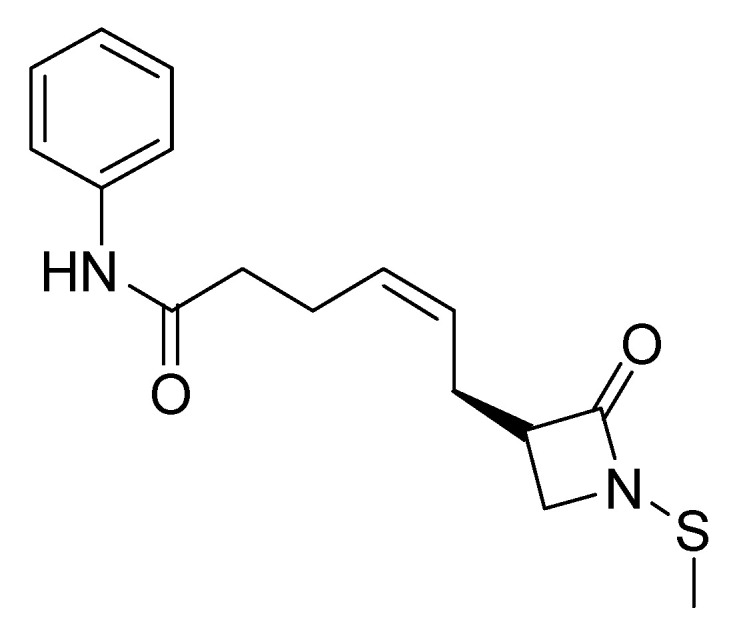
Galletti, P. et al. [180] performed molecular docking studies on (*S*,*Z*)-6-(1-(methylthio)-2-oxoazetidin-3-yl)-*N*-phenylhex-4-enamide in complex with HDAC8.

**Figure 32 pharmaceuticals-16-01639-f032:**
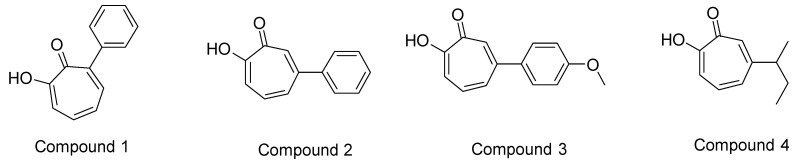
Compounds synthesized by Ononye, S.N. et al. [182] with potent inhibitory activity against HDAC8.

## Data Availability

Not applicable.

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
