# Peer review of "A Review on Molecular Docking on HDAC Isoforms: Novel Tool for Designing Selective Inhibitors"

_pharmaceuticals, 2023, doi:10.3390/ph16121639_

Round 1
Reviewer 1 Report (Previous Reviewer 1)
Comments and Suggestions for Authors
The manuscript is now significantly improved by the addition and discussion of different docking algorithms and programs as well as their limitations.
I recommend to publish this review, if the following minor suggestions are appropriately considered:
The HDAC8 chapter can still be improved. A closer look at this chapter reveals that it covers only less relevant and more or less unselective ligands such as APHA, SAHA, VPA and Tubacin. Urushiol analogs are even less instructive, since there is no experimental data that there is an existent interaction with HDAC8. It appears more than possible that these compounds exert their effect through other target molecules.
On the other side there are well documented ligands with docking data for experimentally validated and selective HDAC8/ligand interactions.
It would be instructive to include docking studies of experimentally validated structurally diverse and non-hydroxamate HDAC8 inhibititors in order to demonstrate the power of docking to predict their binding poses. Informative examples are e.g. the following DOI-numbers:
-10.1021/jm5008209
-10.1021/acs.jmedchem.1c00491
-10.1016/j.bmcl.2015.07.065
-10.1002/cmdc.200900309
-10.1021/ml400158k
Also, it should be discussed, how docking of extraordinarily flexible HDAC8 could create better results by comprehensive mutliple receptor docking as described in DOI: 10.1007/s00894-011-1297-8.
Author Response
Firstly, we would like to thank the reviewer for the fruitful comments.
We have followed the reviewers’ suggestions, aiming to improve our manuscript.
We have included docking studies of experimentally validated structurally diverse and non-hydroxamate HDAC8 inhibitors discussing and citing all the proposed references.
At the conclusion part it is discussed additionally how docking of extraordinarily flexible HDAC8 could create better results by comprehensive multiple receptor docking.
Reviewer 2 Report (Previous Reviewer 3)
Comments and Suggestions for Authors
The manuscript has been notably improved. I thank the authors for considering my suggestions.
Author Response
We really appreciate your help for improving our manuscript.
This manuscript is a resubmission of an earlier submission. The following is a list of the peer review reports and author responses from that submission.
Round 1
Reviewer 1 Report
Comments and Suggestions for Authors
The present paper is a review about docking results on different HDAC isozymes.
The authors claim that molecular docking would be a novel tool for designing selective inhibitors.
First of all, molecular docking is a routine method, which is regularly used to look at potential binding modes of ligands, if
structural data from xray and NMR of protein-ligand complexes for these particular ligands are missing.
The forecasting power of docking is rather limited and depends on the properties of the receptor protein, particularly its flexibilty/malleability.
For example no available docking program can predict the opening of a transient pocket, which enables a completely new binding mode.
This can be partially cured by allowing some mobility around the binding pocket. Also, and this is shown by many publications, which benchmark different docking algorithms and energy scoring functions, the docking results depend greatly from the applied procedure as well as the preparation of the protein 3D-structure. All of these issues and practical considerations connected to molecular docking are not mentioned at all, but are
crucial for the assessment of the value of docking for the development of selective inhibitors.
The review is just an uncritical stringing together of docking results, without any discussion, where docking was really a key technique for the develpment of a successful selective inhibitor or where the limits of docking lie.
Therefore, I cannot see the benefit of this review for users in this field.
For these reasons I cannot recomment to publish this review in Pharmaceuticals.
Other minor issues are:
-replace isoform by isozyme at all locations. Isoforms are splice variants.
-"The action of HAT results in opposite effects and thus maintaining an active balance between the actions of these two enzymes is crucial"
HDAC and HAT are not only two enzymes, but rather represent protein families.
Fig. 2: Chidamide instead of Clidamide; the drawing of Romidepsin appears odd, especially the much larger double bond on the left side.
To my knowledge flavonid are rather unspecific and weak HDAC inhibitors. There is also only poor evidence that the anti-cancer effects are
really linked to HDAC inhibition. Please provide sound evidence that this relationship is existing.
Fig. 7, 10, ..: poor quality.
Docking poses in all figures should be overworked using one program and style to make them uniform.
Author Response
REVIEWER I
First of all we would like to thank the reviewer for the fruitful comments and suggestions aiming to improve the quality of our manuscript.
Comments and Suggestions for Authors
The present paper is a review about docking results on different HDAC isozymes.
The authors claim that molecular docking would be a novel tool for designing selective inhibitors.
First of all, molecular docking is a routine method, which is regularly used to look at potential binding modes of ligands, if structural data from xray and NMR of protein-ligand complexes for these particular ligands are missing. The forecasting power of docking is rather limited and depends on the properties of the receptor protein, particularly its flexibilty/malleability.
Molecular docking is an established in silico structure-based method widely used in drug discovery. Many references highlight its importance since it helps to model the interactions between small molecules and a protein allowing to characterize the behavior of small molecules in the binding site of target proteins as well as to elucidate fundamental biochemical processes. Nowadays, the application of molecular docking has expanded to the prediction of adverse effects, polypharmacology, drug repurposing, and target fishing and profiling [Pinzi L, Rastelli G. Int J Mol Sci. 2019 Sep 4;20(18):4331, doi: 10.3390/ijms20184331].
For example, no available docking program can predict the opening of a transient pocket, which enables a completely new binding mode. This can be partially cured by allowing some mobility around the binding pocket.
We completely agree with the reviewer’s comment. For this issue in most of the published cases flexible docking is performed-allowing the flexibility of the protein and the ligand. Moreover, it has to be noted that for the majority of docking studies MD simulations are performed firstly before proceeding to the docking evaluation of the ligands [Stank, A., et al., Accounts of Chemical Research 2016 49 (5), 809-815, doi: 10.1021/acs.accounts.5b00516, Zheng, X. et al., AAPS J. 2013 Jan;15(1):228-41, doi: 10.1208/s12248-012-9426-6].
This manuscript is a review and deals with the collection of data obtained from the literature.
Also, and this is shown by many publications, which benchmark different docking algorithms and energy scoring functions, the docking results depend greatly from the applied procedure as well as the preparation of the protein 3D-structure. All of these issues and practical considerations connected to molecular docking are not mentioned at all, but are crucial for the assessment of the value of docking for the development of selective inhibitors.
It is well known that depending on the applied docking program different docking algorithms are used and energy scoring functions can be obtained. In this review article the docking scores and procedure are mentioned; additionally the original references have been added in the case that the reader needs more information about each case. It has to be noted that the crucial interactions for inhibition between different novel series of compounds and the amino acids of each isozyme are extensively described. This is important for medicinal chemists for refining the design and improving the selectivity of potential inhibitors.
The review is just an uncritical stringing together of docking results, without any discussion, where docking was really a key technique for the development of a successful selective inhibitor or where the limits of docking lie.
Therefore, I cannot see the benefit of this review for users in this field.
For these reasons I cannot recommend to publish this review in Pharmaceuticals.
As previously mentioned molecular docking is an established tool for modern drug discovery since it highlights the identification of potential drug targets and predicting molecular ligand-target interactions at the atomic level [Sethi, A. Molecular Docking in Modern Drug Discovery: Principles and Recent Applications Drug Discovery and Development - New Advances. IntechOpen; 2020, http://dx.doi.org/10.5772/intechopen.85991, Torres, P.H.M. et al., Int J Mol Sci. 2019 Sep 15;20(18):4574, doi: 10.3390/ijms20184574]. Additionally, application of computational techniques especially molecular docking is essential for drug design in order to reduce the cost and time of bringing a commercial drug to the market [Adelusi, T.I. et al., Informatics in Medicine Unlocked, 29, 2022, 100880]. Different of success stories have been obtained eg. tacrine.
In the present review, data on docking studies of HDAC inhibitors for different compounds’ classes to certain isozymes have been collected from the literature, discussed and conclusions were drawn contributing to the rational design of selective inhibitors.
Other minor issues are:
-replace isoform by isozyme at all locations. Isoforms are splice variants.
Thank you for this suggestion. We have replaced “isoform” by “isozyme”. Thank you for clarifying this difference, although in literature the word “isoform” is commonly used instead of “isozyme” eg. Ganai, S.A.; Farooq, Z.; Banday, S.; Altaf, M. In silico approaches for investigating the binding propensity of apigenin and luteolin against class I HDAC isoforms. Future Med Chem 2018, 10, 1925–1945. 10.4155/fmc-2018-0020.
-"The action of HAT results in opposite effects and thus maintaining an active balance between the actions of these two enzymes is crucial"
HDAC and HAT are not only two enzymes, but rather represent protein families.
This phrase has been corrected.
Fig. 2: Chidamide instead of Clidamide;
We have done the correction.
The drawing of Romidepsin appears odd, especially the much larger double bond on the left side.
We have checked the structure and corrected the large double bond.
To my knowledge flavonoid are rather unspecific and weak HDAC inhibitors.
The purpose of this review is to gather all the literature data concerning the docking studies of HDAC. Scafuri, B. and co-workers [Reference 75] collected already published data of HDAC inhibition [Bontempo, P. et al., Int. J. Biochem. Cell Biol. 2007, 39, 1902–1914, Pandey, M. et al., Mol. Carcinog. 2012, 51, 952–962., and Attoub, S. et al., Eur. J. Pharmacol. 2011, 651, 18–25] to perform molecular docking studies and explore the mechanism on HDAC of flavone and its derivatives apigenin and luteolin. According to these references the activity of these flavones has been already observed by experimental results and reported but it is not known how these molecules bind and inhibit their protein targets.
There is also only poor evidence that the anti-cancer effects are really linked to HDAC inhibition. Please provide sound evidence that this relationship is existing.
As referred in the paper HDAC 1 is related to various cancer types, such as gastric and prostate cancer and seems to exert an influence on the progression of breast cancer by modulating both the expression and transcriptional activity of the estrogen receptor protein alpha [references 67-69]. So there is a relationship between HDAC 1 and cancer. Furthermore, among the inhibitors acting on classes I-II HDACs, are the known anticancer agents Vorinostat and Trichostatin A (TSA) which exert multiple biological effects by interfering with the cell cycle [Reference 75]. Additionally, HDAC are associated with fibrosis and cancer eg. Yoon, S.; Kang, G.; Eom, G.H. HDAC inhibitors: Therapeutic potential in fibrosis-associated human diseases. Int. J. Mol. Sci. 2019, 20, 1329.
Fig. 7, 10: poor quality.
We have tried to improve the quality of these images.
Docking poses in all figures should be overworked using one program and style to make them uniform.
The paper is a review collecting all the data on docking studies of inhibitors on the most important isozymes of HDAC. Each group has used her own program, platform and algorithms. We have reviewed and selected the appropriate material for the review. Τhis is not our research study but a review so we have collected data from previous publications. It was not our goal to perform a docking study using the same program and style to make them uniform.
Reviewer 2 Report
Comments and Suggestions for Authors
The review shows a lot of promise, but some issues need to be addressed mostly the figures. The paper needs to be structured well. Most of the figures need improvement and organized properly with better illustrations. The figures can be merged into four or five. Figures 10 and 27 are blurry. Remove (.) after figure in paragraphs for example (Figure 1.).
Line 19: most critical isoforms of ???
Line 25-26: Reference missing
Line 132 and Line 142: Check the gap between 14 A
Line 136: Check (.)after hydrolysis
Line 149: Protein 1 dimer?
Line 149-183: Suggest adding more references.
Line 203/204: Either vorinostat or just SAHA
Also, the conclusion needs to be improved.
Comments on the Quality of English Language
Minor editing and language editing required.
Author Response
First of all we would like to thank the reviewer for the fruitful comments and suggestions aiming to improve the quality of our manuscript.
REVIEWER II
Comments and Suggestions for Authors
The review shows a lot of promise, but some issues need to be addressed mostly the figures. The paper needs to be structured well. Most of the figures need improvement and organized properly with better illustrations. The figures can be merged into four or five. Figures 10 and 27 are blurry.
Figures 7, 10 and 27 are the same as at the original open access papers. We cannot merge them in 4-5 since different compounds are studied with different programs (results obtained from various laboratories).
Remove (.) after figure in paragraphs for example (Figure 1.).
The (.) has been removed.
Line 19: most critical isoforms of ???
It has been corrected.
Line 25-26: Reference missing
References have been added.
Line 132 and Line 142: Check the gap between 14 A
We have corrected this gap.
Line 136: Check (.) after hydrolysis
(.) has been removed.
Line 149: Protein 1 dimer?
We have added a phrase explaining the protein 1 dimer.
Line 149-183: Suggest adding more references.
More references explaining this case have been added.
Line 203/204: Either vorinostat or just SAHA
Thank you we have kept the name “vorinostat”. SAHA (suberoylanilide hydroxamic acid) appears only at the beginning at line 66 to explain that it is the same as vorinostat.
Also, the conclusion needs to be improved.
We have improved the conclusions.
Comments on the Quality of English Language. Minor editing and language editing required.
The manuscript has been checked by a native speaker and english have been improved.
Reviewer 3 Report
Comments and Suggestions for Authors
The review presented by Drakontaeidi & Pontiki focuses in specificity and selectivity between HDAC isoforms. This field is of high interest and the overall quality of the review is notable. Here are some suggestions that may improve the manuscript.
Minor comments
Overall, use the term pleiotropic instead of side effects; as not all are negative or plainly unknown.
For Figure 2, please consider an alternative representation for romidepsin.
Line 108 is wrongly placed, between Figure 4 and its label.
Typographical error at the end of line 216.
Missing symbol in line 220.
Major comments:
Please keep the format and style of molecular structures consistent in all the manuscript.
Similarly, there are several structures with distorted bonds.
Some of the discussed compounds are known pan-assay interference compounds, which makes it hard to evaluate the merits of modeling for such cases.
As stated, much of the manuscript focuses on molecular docking implementations; which makes me wonder if there aren't any reports of other methods such as FEP or ligand-based methodologies aiming for selectivity.
A major suggestion I have is the inclusion of any experimental findings that reinforce or validate the presented hypotheses for each isoform.
Comments on the Quality of English LanguageThere are minor errors in text, a revision is advised.
Author Response
Firstly we would like to thank the reviewer for the fruitful comments and suggestions aiming to improve the quality of our manuscript.
REVIEWER III
Comments and Suggestions for Authors
The review presented by Drakontaeidi & Pontiki focuses in specificity and selectivity between HDAC isoforms. This field is of high interest and the overall quality of the review is notable. Here are some suggestions that may improve the manuscript.
Minor comments
Overall, use the term pleiotropic instead of side effects; as not all are negative or plainly unknown.
Thank you for this suggestion. We have corrected this term.
For Figure 2, please consider an alternative representation for romidepsin.
We have completely changed this figure.
Line 108 is wrongly placed, between Figure 4 and its label.
We have corrected the placement of Line 108.
Typographical error at the end of line 216.
It has been corrected.
Missing symbol in line 220.
The symbol has been added.
Major comments:
Please keep the format and style of molecular structures consistent in all the manuscript.
Similarly, there are several structures with distorted bonds.
We have checked and corrected all the structures.
Some of the discussed compounds are known pan-assay interference compounds, which makes it hard to evaluate the merits of modeling for such cases.
It is true that pan-assay interference compounds (PAINS) often give false positive results in high-throughput screens and react nonspecifically with numerous biological targets rather than specifically affecting one desired target. All the compounds mentioned in this review have been biologically evaluated and studied for their binding mode to the specific isozyme as studied. Docking studies provide information about the interactions of the compounds with the amino acids of the active center.
As stated, much of the manuscript focuses on molecular docking implementations, which makes me wonder if there aren't any reports of other methods such as FEP or ligand-based methodologies aiming for selectivity.
There are only few studies of other methods such as FEP or ligand-based methodologies aiming for selectivity. For example, in reference 108 a computational approach is applied to predict the selectivity profile of developed inhibitors. Molecular docking followed by MD simulation and calculation of binding free energy was performed for a dataset of 2-aminobenzamides comprising 30 previously developed inhibitors. Additionally, in reference 111 different ligand-based and structure-based drug methods have been analyzed to predict the binding mode and inhibitory activity of recently developed alkylhydrazide HDAC inhibitors. Pharmacophore models and atom-based quantitative structure-activity relationship (QSAR) models were generated and evaluated. The binding mode of the studied compounds was determined using molecular docking as well as molecular dynamics simulations and compared with known crystal structures.
In this review we have focused on docking studies of the most important isozymes that is why the other available data are not included.
A major suggestion I have is the inclusion of any experimental findings that reinforce or validate the presented hypotheses for each isoform.
All the experimental data that reinforce or validate the presented hypotheses for each isozyme are included in the references of the original research papers where biological evaluation and docking studies are conducted that is why they have not been added to this review for docking studies. This review is focused on collecting data from the literature for docking studies of HDAC inhibitors. Our aim was, if possible, to highlight the most important interactions between the potential inhibitors and HDAC isozymes, elucidating the necessary structural features that potential therapeutic compounds should possess in order to enhance their efficacy.
Comments on the Quality of English Language. There are minor errors in text, a revision is advised.
The manuscript has been checked by a native speaker and English language have been improved.
Round 2
Reviewer 1 Report
Comments and Suggestions for Authors
My first recommendation was not to publish this manuscript.
Main reasons:
The review is just an uncritical stringing together of docking results and does not contain an adequate discussion about different docking algorithms and their limitations. Moreover, the possibilities of docking are clearly overestimated. There are success stories, but also cases where docking does not help at all. This critical discussion of docking is completely lacking in my opinion.
Therefore, in my eyes, the improvements are not enough to justify publication.